



# Reconstructing sea level rise at global 945 tide gauges since 1900

Dapeng Mu[1], Ruhui Huang[2], Peng Yin[1], Haoming Yan[3], Tianhe Xu[1]

[1]Institute of Space Research, Shandong University, Weihai, 264209, China
[2]State Key Laboratory of Marine Environmental Science, Center for Marine Meteorology and Climate Change, College of Ocean and Earth Sciences, Xiamen University, Xiamen, 361102, China.
[3]Innovation Academy for Precision Measurement Science and Technology, Chinese Academy of Science, Wuhan, 430077, China.

*Correspondence to*: Tianhe Xu (thxu@sdu.edu.cn)

**Abstract.** Tide gauges record sea level changes along coast. They are widely used to determine the twentieth century global mean sea level (GMSL) rise. However, a major issue in tide gauge data is the presence of various, substantial, and sometime persistent data gaps, which hinder our understanding of sea level rise, especially at regional and local scales. Whilst the GMSL reconstructions have been provided by several influential studies, reconstructions at the exact sites of tide gauges are rarely available. Here, we present sea level reconstructions at global 945 tide gauges, covering the period over 1900 to 2022. Our approach relies on a data assimilation technique that integrates various physical sea level observations and predictions, including sea level simulations from 35 climate models. A prominent feature in our reconstruction is that it provides an ensemble of 35 reconstructions at each site of tide gauge, offering complete and refined sea level time series. This ensemble reconstruction allows for direct statistical assessment, e.g., average, median, spread, and percentile. The average of reconstructed sea level across 945 tide gauges reveals a GMSL rise of 1.75±0.05 mm/yr over 1900-2020, and shows strong agreements with other GMSL reconstructions for both the curves of time series and overall trends. At local scale, our reconstructions are comparable to an independent reconstruction, despite apparent rate differences at locations, it is suggested that our reconstructed sea level trends closely follow the raw records when they are available, emphasizing the importance of the observed sea level rise at tide gauges. Our sea level reconstructions offer a valuable resource for improving global and regional sea level projections, validating climate model performance, and informing coastal adaptation strategies. The reconstructed sea level is available at https://doi.org/10.5281/zenodo.15385035. (Mu, 2025).

## 1 Introduction

Tide gauges sample relative sea level changes along coast. The longest records date back to the early nineteenth century (Figure 1a), according to the data collection by the Permanent Service for Mean Sea Level (PSMSL) website (https://psmsl.org/). Records of tide gauges are widely applied to geoscientific investigations. Extensive applications include estimating long-term sea level rise and acceleration (Douglas, 1991; Holgate, 2007; Woodworth et al., 2009); determining vertical land motion in combination with satellite altimetry (Woppelmann & Marcos, 2016; Zhou et al., 2022; Oelsmann et al., 2024); investigating the oceanic response to atmospheric loading (Ponte, 2006; Piecuch & Ponte, 2015; Zhu et al., 2024);

assessing wind-driven variability along coasts (Thompson et al., 2014; Little, 2023); evaluating extreme sea level events across various time scales (Calafat et al., 2022; Moftakhari et al., 2024); examining interaction with climate variability (Kenigson et al., 2018; Royston et al., 2022); and identifying the contributing sources to sea level rise (Frederikse et al., 2016;

Calafat et al., 2022; Mu et al., 2024a). These studies highlight the essential role of tide gauges in advancing our understanding sea level changes in response to climate change.

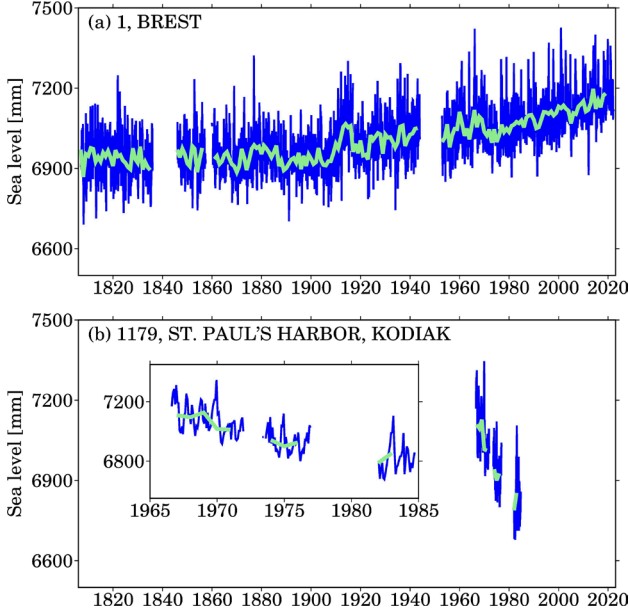

**Figure 1: Examples of tide gauges from PSMSL; (a) Brest, ID 1, France; (b) St Paul's Harbor, Kodiak, ID 1179, USA; blue lines are monthly records and the light green lines are annual records.**

However, a notorious issue in the records of tide gauges is the data gaps (Piecuch et al., 2017). Typical gaps are characterized with substantial discontinuity, e.g., a blank over more than decades, or a very short duration of years only, (see Figure 1b). These data gaps may result from the site maintenance issues, instrument destruction, and complete submergence by high sea level owing to strong climate variability, such as the 1997/1998 El Niño event. Regardless of their cause, such data gaps impede applications of tide gauges in long-term sea level related studies. To address this challenge, various sea

level reconstruction approaches are proposed to fill the data gaps in the tide gauge records. Classic reconstruction approaches involve empirical orthogonal function (EOF) reconstruction (Chambers et al., 2000; Church et al., 2004), and data assimilation technique (Hay et al., 2013; Calafat et al., 2022; Mu et al., 2024b). Other strategies include station stacking (Jevrejeva et al., 2014), as well as spatial and temporal interpolation or extrapolation using neural networks (Wenzel & Schröter, 2011) and Bayesian inference (Choblet et al., 2014; Piecuch et al., 2017).

The EOF reconstruction has been widely applied to sea level reconstruction. This approach extracts basic functions from satellite altimetry (Church et al., 2004; Church & White, 2011) or model simulations (Berge-Nguyen et al., 2008). The dominant modes (i.e., basic functions) are then combined with tide gauges to determine the amplitudes of those modes in



least-square manners (Ray & Douglas, 2011). The prominent advantage of EOF reconstruction is that it produces sea level reconstruction fields with near-global coverage (matching satellite altimetry or model domains) with extension to the whole period of tide gauge records. Since the observational fields from satellite altimetry are usually removed with trends and seasonal cycles, traditional EOF reconstruction is only able to resolve the variability in sea level (Chambers et al., 2000), but not to capture the long-term trends. To address this issue, Church et al. (2004) proposed the EOF0 mode (i.e., values of ones are filled in this mode) and added it into the basic functions extracted from satellite altimetry. The resulting reconstruction retrieves the long-term trends in global sea level rise. A theoretical exploration on EOF reconstruction, especially the EOF0 component, was presented by Calafat et al. (2014). They found that, by nature, the EOF reconstruction is a weighting scheme for tide gauges. It is also should be noted that the EOF reconstruction recovers sea level changes at the predefined grids (e.g., the satellite altimetry product grids or an ocean model grid), it does not produce direct estimates at tide gauges.

A variant approach of EOF reconstruction is the cyclostationary EOF (CSEOF) reconstruction, which was developed by Hamlington et al. (2011). In contrast to the stationary basic functions from EOF, CSEOF acquires non-stationary basic functions that better describe annual cycles and some major climate variability such like the El Niño–Southern Oscillation (ENSO) (Hamlington et al., 2015). Therefore, the CSEOF reconstruction is capable of recovering non-stationary spatial variability due to ENSO, in addition to sea level rise.

Data assimilation provides another powerful framework for sea level reconstruction (Hay et al., 2013, 2015). In contrast to EOF or CSEOF methods, which are mathematically driven, data assimilation relies on physical orientated basic functions, filling the data gaps with physically meaningful interpolation/extrapolation (Mu et al., 2024a). These basic functions either describe redistributions of water mass exchange between land and oceans (Tamisiea, 2011) or represent changes in sea level due to steric effect and circulations (Gregory et al., 2019; Huang et al., 2025). The data assimilation technique was first proposed by Hay et al. (2013) in a simulation study and later applied to reconstruct the twentieth century sea level rise with 622 tide gauges. Mu et al. (2024b) modified this approach with a focus on a regional case (China coast). Their GMSL reconstruction aligns with other GMSL reconstructions. Calafat et al. (2022) developed a different type of data assimilation to reconstruct sea level rise in the Mediterranean Sea Since 1960, along with its contributing sources. Beyond reconstruction, data assimilation technique also permits for inferring ocean mass increase (Mu et al., 2024a) and sterodynamic sea level changes (Calafat et al., 2022). The data assimilation approach can be applied to either tide gauges, emphasizing individual, local changes, or to a two-dimension field, such as satellite altimetry, resolving spatial variability (Dangendorf et al., 2024).

The spatial or temporal interpolation/extrapolation approach is implemented through several techniques. Wenzel & Schröter (2010, 2014) presented sea level reconstruction with neural networks. Their method was constructed through training the neural networks with data from satellite altimetry or a reconstructed field by EOF reconstruction. A notable feature of their approach is that it is applied to monthly records rather than annual records, which thus allows the recovery of high-frequency variability. Piecuch et al. (2017) introduced a Bayesian algorithm for sea level reconstruction, and its fully Bayesian version accounts for uncertainty in model parameters. Both these studies focus on temporal interpolation/extrapolation, highlighting sea level changes at tide gauges. A distinct version of the Bayesian inference for sea





level reconstruction is the trans-dimensional regression (Hawkins et al., 2019), which performs spatial interpolation/extrapolation on rates of sea level rise, rather than sea level time series. This method parameterizes Earth's surface using various structures associated with prescribed probability density functions, and generates either spatially
continuous grids or specific coastal grids covering global coast (e.g., Oelsmann et al., 2024).

A conceptually straight method for sea level reconstruction is virtual station stacking (Jevrejeva et al., 2014), which merges the two closest tide gauges into a single "virtual" station and iterates this process until the virtual station converges to a final, unique station over the globe or for a given region (e.g., Pacific). Readers can see the Figure 5 from Grinsted et al. (2007) for a direct illustration. This method creates the longest records for sea level reconstruction that dates back to 1807,
and also allows for examination for reginal sea level rise and acceleration (Jevrejeva et al., 2014). However, the station stacking method only permits regional or global sea level reconstruction, it does not reconstruct sea level time series at tide gauges, because this method does not create interpolations or extrapolations.

To date, several distinguished literatures (e.g., Church & White 2011; Ray & Douglas, 2011; Jevrejeva et al., 2014; Hay et al., 2015; Dangendorf et al., 2019; Frederikse et al., 2020) have already released their GMSL reconstructions to the
community. These GMSL curves have been extensively applied to a range of sea level and climate studies, generating profound influence. Treu et al. (2024) released a regional sea level reconstruction whose grid covers global coast. However, this reconstruction was not performed at the exact sites of tide gauge. It involves projection from tide gauges onto satellite altimetry grids (Dangendorf et al., 2019), and spatial interpolations/extrapolations. In this study, we improve the data assimilation method (Hay et al., 2015; Mu et al., 2024a), and use to reconstruct annual sea level changes at the exact sites of
global 945 tide gauge from 1900 to 2022. Furthermore, instead of a single reconstruction time series, we offer an ensemble of reconstructions that include 35 complete time series for each tide gauges. The resulting complete records will provide valuable inputs for regional assessments of the twentieth century sea level rise, especially at regional and local scales.

## 2 Methods and data

### 2.1 Sea level reconstruction by data assimilation

In this subsection, we outline the implementation of the data assimilation approach. We begin by introducing the basic concept to facilitate understanding for readers, followed by a detailed description of the computational procedures. The data assimilation approach consists of two fundamental stages. In the first stage, observation equations are constructed using raw tide gauge records. In the second stage, physically orientated processes are prescribed to represent the relative sea level rise at tide gauges (Frederikse et al., 2020; Calafat et al., 2022). These processes involve three major mechanisms. The first one
is the sea level changes resulting from the ocean circulations and steric effect, which is also referred to as sterodynamic sea level (SDSL) changes (Gregory et al., 2019). We utilize outputs from the Coupled Model Intercomparison Project Phase 6 (CMIP6) climate models to represent SDSL changes at tide gauges (see subsection 2.4). However, due to model configurations, SDSL does not account for global mean ocean mass changes (Griffies et al., 2016). To account for changes



in ocean mass, we further introduce the second process that depicts the water mass exchange between oceans and land,
including Greenland ice melting, Antarctica ice melting, mountain glacier melting, and terrestrial water storage variations
(Gregory et al., 2013). These contributions redistribute over oceans and form unique geometries under the gravity, rotation,
and deformation (GRD) effect (Mitrovica et al., 2011; Coulson et al., 2022). Those oceanic geometries are termed as sea
level fingerprint (SLF; Coulson et al., 2022). A random process is further proposed to account for model deficiencies at local
scale, because climate models tend to underestimate the sea level changes (Meyssicnac et al., 2017). The final mechanism
reflects the ongoing effect from glacial isostatic adjustment (GIA) (Peltier et al., 2015), which influences the relative sea
level measured by tide gauges. The three processes constitute the relative sea level rise along coast, and they have global
physical origins. We therefore express the increment in sea level at tide gauges ($\Delta\mathcal{SL}$) in mathematical form:

$$\Delta\mathcal{SL} = \dot{\mathcal{SL}}(t)_{SLF}\Delta t + \dot{\mathcal{SL}}(t)_{SDSL}\Delta t + \dot{\mathcal{SL}}(t)_{GIA}\Delta t \tag{1}$$

where $\dot{\mathcal{SL}}_{GIA}$ is the rate of GIA relative sea level; $\dot{\mathcal{SL}}(t)_{SDSL}$ is the rate of SDSL; $\dot{\mathcal{SL}}(t)_{SLF}$ is the rate of SLF,
representing ocean mass increase.

Although the CMIP6 climate models provide SDSL estimates, they may not accurately capture local variations at tide
gauge sites. To better address these local changes, we introduce a random process. Therefore, the $\dot{\mathcal{SL}}(t)_{SDSL}$ include two
parts:

$$\dot{\mathcal{SL}}(t)_{SDSL} = \dot{\mathcal{SL}}(t)_{Random} + \dot{\mathcal{SL}}(t)_{Model} \tag{2}$$

Where $\dot{\mathcal{SL}}(t)_{Model}$ is the SDSL output simulated by CMIP6 climate models, while $\dot{\mathcal{SL}}(t)_{Random}$ is unknown variable, and
will be estimated by our data assimilation framework.

Combining equations (1) and (2), at a given tide gauge, its sea level at time $t+1$ ($\mathcal{SL}^{t+1}$) can be evolved from sea
level at time $t$ ($\mathcal{SL}^{t}$):

$$\mathcal{SL}^{t+1} = \mathcal{SL}^{t} + \dot{\mathcal{SL}}(t)_{SLF}\Delta t + \dot{\mathcal{SL}}(t)_{Random}\Delta t + \dot{\mathcal{SL}}(t)_{Model}\Delta t + \dot{\mathcal{SL}}(t)_{GIA}\Delta t \tag{3}$$

Equation (3) essentially describes how sea level rise evolves over time, or defines how sea level rise transitions from time $t$
into time $t+1$. This equation contains two roles: the first one involves known variables, i.e., $\dot{\mathcal{SL}}(t)_{GIA}$ and $\dot{\mathcal{SL}}(t)_{Model}$,
which act as the 'model driven' role; the second one involves variables to be estimated through data assimilation, i.e.,



$\dot{\mathcal{SL}}(t), \dot{\mathcal{SL}}(t)_{Random}$, and the amplitude of $\dot{\mathcal{SL}}(t)_{SLF}$. We stress that $\dot{\mathcal{SL}}_{SLF}$ is the overall trend of sea level fingerprint since 1900, and it is different from $\dot{\mathcal{SL}}(t)_{SLF}$, which represents the rate of sea level fingerprint at time step $t$, their

mathematical relation is:

$$\dot{\mathcal{SL}}(t)_{SLF} = \alpha(t) \times \dot{\mathcal{SL}}_{SLF} \tag{4}$$

where $\alpha$ is the amplitude of $\dot{\mathcal{SL}}_{SLF}$ at time step $t$. This equation implies that that their amplitudes is time variable, but the spatial pattern is fixed.

We use $X^t$ to represent the state vector. At every time step $t$, the observation equation is defined as:


$$Z^t = H^t X^t + \varepsilon \tag{5}$$

where $Z^t$ is the observational vector containing sea level records from the selected tide gauges (subsection 2.3) at time $t$. Its dimension is time variable, and equals to the available number ($m$) of tide gauges (Figure 2). $\varepsilon$ denotes observational noise, and $H^t$ is the mapping matrix, consisting of two parts:

$$H^t = \begin{bmatrix} H^t_{TG} & H^t_{other} \end{bmatrix} \tag{6}$$

where $H^t_{other} = 0_{m \times (n+1)}$, $n$ is the total number of tide gauges selected (for example, 945 tide gauges selected by this paper, see subsection 2.3); $H^t_{TG}$ is sparse matrix, for each row, the $i$th element is one if the $i$th tide gauge record is available, otherwise, it is zero; the dimension of $H^t_{TG}$ is $m \times n$.

In our data assimilation, the state vector $X^t$ contains:

$$X^t = \begin{bmatrix} \mathcal{SL}^t_1 \\ \vdots \\ \mathcal{SL}^t_n \\ \dot{\mathcal{SL}}^t_{Random,1} \\ \vdots \\ \dot{\mathcal{SL}}^t_{Random,n} \\ \alpha^t \end{bmatrix} \tag{7}$$





where $\mathcal{SL}_i^t$ represents sea level at $i$th tide gauge at time $t$; $\dot{\mathcal{SL}}_{Random,i}^t$ represents rate of random sea level processes at $i$th

tide gauge; $\alpha^t$ is the amplitude of sea level fingerprint at time $t$. In the filter, the state transition matrix $\Phi$ transforms the

state vector into next time step:

$$X_f^{t+1} = \Phi X_a^t + \dot{\mathcal{SL}}(t)_{Model}\Delta t + \dot{\mathcal{SL}}(t)_{GIA}\Delta t + w \qquad (8)$$

where $w$ represents the model noise, subscript $f$ denotes the 'forecast' state, while subscript $a$ denotes 'analysis' solution.

The computation of $\dot{\mathcal{SL}}(t)_{Model}$ is detailed in the next subsection. $\dot{\mathcal{SL}}(t)_{Model}$ and $\dot{\mathcal{SL}}(t)_{GIA}$ serve as a driven role in our

data assimilation framework and are not part of the state vector. The state transition matrix $\Phi$ is constructed as follows:

$$\Phi = \begin{bmatrix} I_{n\times n} & I_{n\times n} & y_{SLF} \\ 0_{n\times n} & I_{n\times n} & 0_{n\times 1} \\ 0_{1\times n} & 0_{1\times n} & 1 \end{bmatrix} \qquad (9)$$

where $y_{SLF}$ contains $\dot{\mathcal{SL}}_{SLF}$ at tide gauges, and its dimension is $n\times 1$.

Equations (5) and (8) constitute the primary formulism of our data assimilation scheme:

$$Z^t = H^t X^t + \varepsilon, \varepsilon \sim N(0,R)$$

170                                                   (10)

$$X_f^{t+1} = \Phi X_a^t + \dot{\mathcal{SL}}(t)_{Model}\Delta t + \dot{\mathcal{SL}}(t)_{GIA}\Delta t + w, w \sim N(0,Q)$$

where $R$ denotes observation noise, and $Q$ represents the covariance matrix of the state vector variables. The covariance

structure is given by:

$$Q = \begin{bmatrix} V_{TG} & 0 \\ 0 & \sigma^2 I_{(n+1)\times(n+1)} \end{bmatrix} \qquad (11)$$

where $I_{(n+1)\times(n+1)}$ is the identity matrix, implying that there are no correlations among random processes at tide gauges,

because we assume random processes sea level and the sea level fingerprint are independent. $\sigma^2$ is a parameter that defines

how much $\dot{\mathcal{SL}}_{Random,i}^t$ and $\alpha(t)$ can vary over time, in our practice, we set its value to be 1 mm/yr; $V_{TG}$ defines

correlation between tide gauges, and it is computed using the distance between tide gauges:



$$V_{TG} = \sigma_{TG}^2 \left( e^{\left( \frac{-D}{\tau} \right)} - 0.4 \right) \tag{12}$$

where $\sigma_{TG}^2$ is the variance of detrended tide gauges; $\tau$ is the decorrelation length scale, which is assumed to be 500 km; and $D$ is the distance between tide gauges. Correlation is only considered for pairs with $D \leq 300$ km, otherwise, tide gauges are treated as uncorrelated.

The data assimilation can be solved recursively using following equations:

$$v_t = Z^t - H^t X^t, F_t = H^t P_t H^{t\prime} + R$$
$$X_a^t = X^t + P_t H^{t\prime} F_t^{-1} v^t, P_{aa}^t = P_f^t - P_f^t H^{t\prime} F_t^{-1} H^t P_f^t \tag{12}$$
$$X_f^{t+1} = \Phi X_f^t + K_t v_t, P_f^{t+1} = \Phi P_f^t (\Phi - K_t H^t)' + Q$$

where $K_t = \Phi P_f^t H^{t\prime} F_t^{-1}$ is the Kalman gain matrix, and $v_t$ is the innovation with variance $F_t$ (Didova et al., 2016).

In the smoother (i.e., the backward loop) process, the Kalman smoother comprises the equations:

$$r_{t-1} = H^{t\prime} F_t^{-1} v_t + L_t' r_t, N_{t-1} = H^{t\prime} F_t^{-1} H^t + L_t' N_t L_t$$
$$\widehat{X}^t = X^t + P_t r_{t-1}, V_t = P_t - P_t N_{t-1} P_t \tag{13}$$

where $L_t = \Phi - K_t H^t$, and $\widehat{X}^t$ represents the smoothed state vector.

**2.2 Instantaneous rate of SDSL changes**

The Instantaneous rate of SDSL changes $\dot{\mathcal{SL}}(t)_{Model}$ acts as drivers in our data assimilation framework. Two essential computational steps are employed to determine $\dot{\mathcal{SL}}(t)_{Model}$. First, we extract the low-frequency variation $x(t)$ from a given SDSL time series simulated by CMIP6 climate models. Second, we estimate the instantaneous rate by computing the first-order temporal derivative of $x(t)$.

Given a raw time series $y(t)$ with unit of millimetre, we apply Hodrick-Prescott (HP) filtering (Kim et al., 2009) to extract its low-frequency variation:

$$x^{HP} = \left( I + 2\lambda T'T \right)^{-1} y \tag{14}$$

where $T$ is Toeplitz matrix:



$$T = \begin{bmatrix} 1 & -2 & 1 & & & \\ & 1 & -2 & 1 & & \\ & & \ddots & \ddots & \ddots & \\ & & & 1 & -2 & 1 \\ & & & & 1 & -2 & 1 \end{bmatrix} \qquad (15)$$

Based on the $x^{HP}$, we estimate the instantaneous rate $\dot{\mathcal{SL}}(t)_{Model}$ by taking the first-order temporal derivative. Since clime model historical outputs cover the period 1900-2014, the instantaneous rate is only available for this period. For the extension period of 2015–2022, we assume that the instantaneous rates remain the same rate as the year of 2014, and construct a complete time series for 1900-2022.

**2.3 Tide gauges**

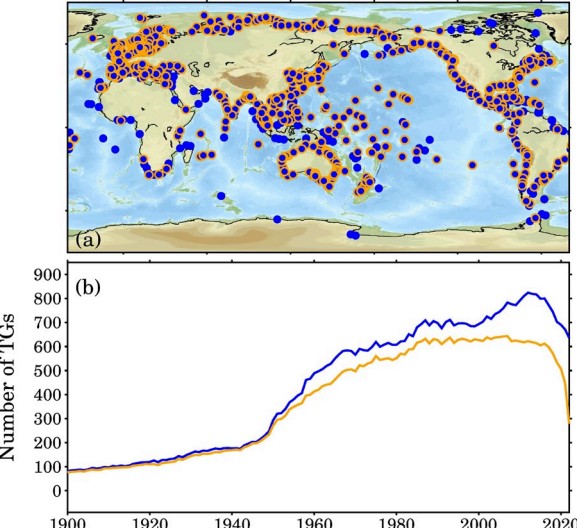

**Figure 2: Tide gauges from the PSMSL (access on 25 January 2025). (a) Distribution of all tide gauges, and those marked with orange circles are selected in this study. (b) The available numbers of tide gauges: the blue line represents all available tide gauges, while the orange line includes only the selected subset. The apparent decline toward the end of the record is primarily due to delays in data updates.**

We consider annual records of tide gauges collected by the PSMSL website (https://psmsl.org/; Holgate et al, 2013). The PSMSL database stored more than 1500 tide gauges that are distributed along the global coastline (Figure 2; data access on 26 November 2024). However, many tide gauge records exhibit substantial data gaps, suspicious anomalies, or abrupt jumps (e.g., Piecuch et al., 2017; Oelsmann et al., 2024). Previous studies (Church & White, 2011; Ray & Douglas, 2011;





Jevrejeva et al., 2014; Hay et al., 2015; Wang et al., 2024; Mu et al., 2024a) have applied various selection criteria to identify reliable tide gauges based on specific research objectives.

In this study, we adopt a single primary criterion: tide gauges must have at least 20 years of data within the period 1900–2022. We do not exclude records with large jumps or high rates, as their impact on global sea-level reconstruction is negligible. After applying this criterion, 945 tide gauges are retained. Figure 2b shows the number of available records for every year over 1900–2022. The orange line represents the records selected by this study. Notably, most pre-1950 records are included, although their number is relatively small—fewer than 300 in total and fewer than 100 during 1900–1910. Over 1900–2022, these 945 tide gauges could potentially provide 116,235 (945×123) data records. However, due to data gaps, only 45,682 records are available, including anomalous records, accounting for only 39.3% completeness over all. Note that the completeness is time variable (see Figure 2b), it is even worse before 1950.

## 2.4 Climate models

**Table 1. 35 CMIP6 models.**

| Index | Model | Experiment | Index | Model | Experiment |
|---|---|---|---|---|---|
| 1 | ACCESS-CM2 | r1i1p1f1 | 19 | GISS-E2-2-G | r1i1p1f1 |
| 2 | ACCESS-ESM1-5 | r1i1p1f1 | 20 | GISS-E2-2-H | r1i1p1f1 |
| 3 | BCC-ESM1 | r1i1p1f1 | 21 | HadGEM3-GC31-MM | r1i1p1f3 |
| 4 | CanESM5 | r1i1p1f1 | 22 | HadGEM3-GC31-LL | r1i1p1f3 |
| 5 | CanESM5-1 | r1i1p1f1 | 23 | INM-CM4-8 | r1i1p1f1 |
| 6 | CanESM5-CanOE | r1i1p2f1 | 24 | INM-CM5-0 | r1i1p1f1 |
| 7 | CMCC-CM2-HR4 | r1i1p1f1 | 25 | IPSL-CM6A-LR | r1i1p1f1 |
| 8 | CMCC-CM2-SR5 | r1i1p1f1 | 26 | IPSL-CM6A-LR-INCA | r1i1p1f1 |
| 9 | CMCC-ESM2 | r1i1p1f1 | 27 | MIROC6 | r1i1p1f1 |
| 10 | CNRM-CM6-1 | r1i1p1f2 | 28 | MPI-ESM-1-2-HR | r1i1p1f1 |
| 11 | CNRM-ESM2-1 | r1i1p1f2 | 29 | MPI-ESM-1-2-LR | r1i1p1f1 |
| 12 | EC-Earth3 | r1i1p1f1 | 30 | MPI-ESM-1-2-HAM | r1i1p1f1 |
| 13 | EC-Earth3-AerChem | r1i1p1f1 | 31 | MRI-ESM2-0 | r1i1p1f1 |
| 14 | EC-Earth3-CC | r1i1p1f1 | 32 | NorESM2-LM | r1i1p1f1 |
| 15 | EC-Earth3-Veg | r1i1p1f1 | 33 | NorESM2-MM | r1i1p1f1 |
| 16 | EC-Earth3-Veg-LR | r1i1p1f1 | 34 | UKESM1-0-LL | r1i1p1f2 |
| 17 | GISS-E2-1-G | r1i1p1f1 | 35 | UKESM1-1-LL | r1i1p1f2 |
| 18 | GISS-E2-1-G-CC | r1i1p1f1 | | | |

In our assimilation framework, we have adopted the SDSL changes estimated by the CMIP6 climate models (Griffies et al., 2016). The SDSL changes describe fluctuations that are attributable to ocean dynamics. This diagnostic field is expected to have a zero global mean (Gregory et al., 2019). Therefore, it does not capture the component of GMSL changes



contributed by the ocean mass increase due to polar ice melting and terrestrial water storage variations. Totally, we include 35 CMIP6 climate models that provide both monthly gridded SDSL fields and global mean thermosteric sea level changes (Table 1).

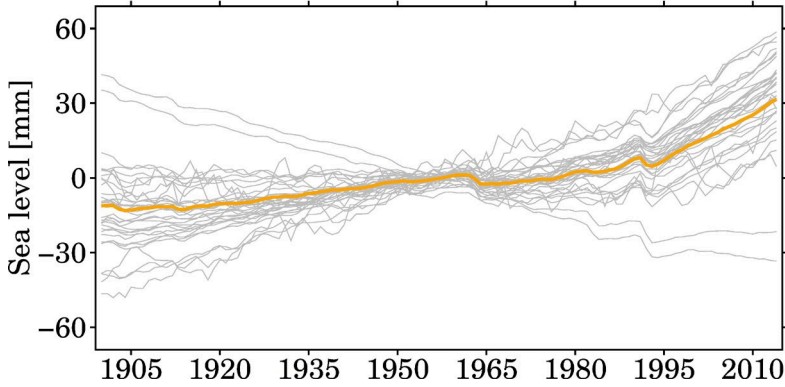


**Figure 3: Global mean thermosteric sea level rise from CMIP6 climate models. Gray lines indicate individual results from 35 climate models, and orange line indicates their ensemble mean.**

## 2.5 Sea level fingerprints

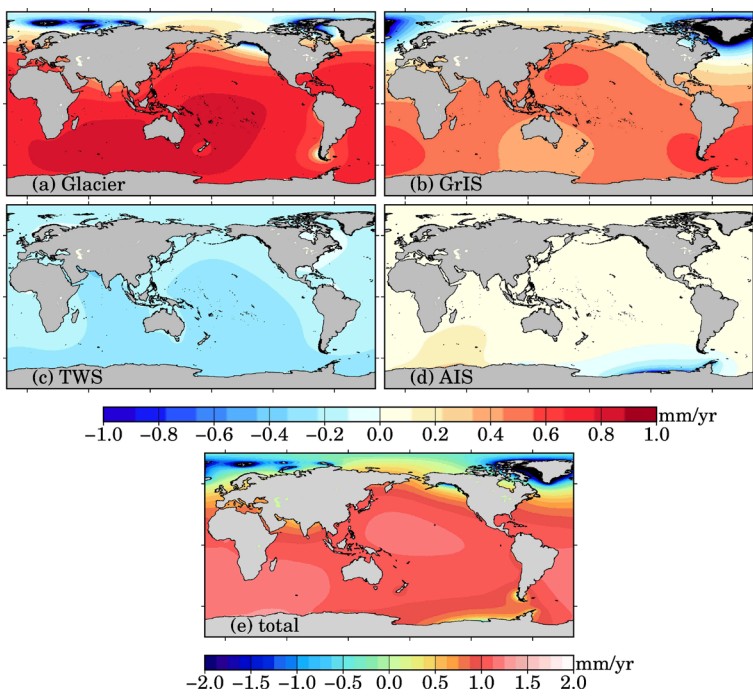

**Figure 4: Sea level fingerprints caused by (a) global mountain glacier, (b) Greenland Ice Sheet, (c) terrestrial water storage variations, and (d) Antarctic Ice Sheet. I The total sea level fingerprints.**

Water mass exchanges between land and oceans involve four major processes: (1) the mass loss or gain in global glaciers; (2) the mass loss from the Greenland Ice Sheet; (3) the mass loss from the Antarctica Ice Sheet; and (4) variations

in terrestrial water storage, driven by both internal nature variability and external anthropogenic forcing. When additional

waters from these sources enters the ocean, it inevitably contributes to global sea level rises. However, this rise is not

spatially uniform. Instead, it exhibits a distinct spatial pattern due to the combined effects of GRD (Farrell & Clark, 1976; Mitrovica et al., 2011; Adhikari et al., 2016). The resulting spatial pattern is known as the SLF, which characterizes the Earth's response to surface mass loading redistribution. Given the centennial timescale considered in this study, we focus solely on the Earth's elastic response. We adopt the total SLF provided by Frederikse et al. (2020), which represent the

integrated contributions from the four global land-based mass redistribution processes (Figure 4). These individual SLF processes are gridded into 0.5° grid, covering the period from 1900 to 2018. We use this dataset to estimate the overall long-term SLF trend.

## 2.6 Glacial isostatic adjustment

The data assimilation technique in this study also requires relative sea level rise contributed from GIA effect. For this,

we use model outputs from ICE-6G-C model (Peltier et al., 2015). Figure 5 shows the spatial pattern of the relative sea level rise from the ICE-6G-C model. The GIA-induced relative sea level is assumed to be purely linear changes for the period from 1900 to 2022, as the GIA-induced relative sea level mainly is an ongoing response to the tremendous ice melting since the Last Glacial Maximum (Calark et al., 2009). In addition, GIA effect also induces an uplift change, i.e., vertical land motion (Hamlington et al., 2016; Woppelmann & Marcos, 2016; Santamaría-Gómez et al., 2017). The GIA-induced relative

sea level rates at all 945 tide gauges are also included in our data files, see section 'Code and data availability'.

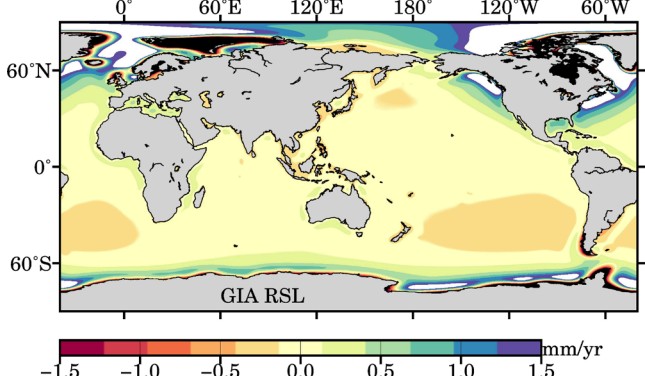

**Figure 5: Rate of relative sea level predicted by the model ICE-6G_C (Peltier et al., 2015).**

## 2.7 Sea level reconstructions from other studies

In this study, our sea level reconstructions are evaluated against publicly available GMSL reconstructions that have

exerted broad influence in sea level studies. These community-accessible GMSL reconstructions are based on various approaches and incorporate different considerations of tide gauges (Table 2). Below is a brief description of each approach. Both Church & White (2011) [C2011] and Ray & Douglas (2011) [R2011] employed the classic EOF reconstruction





technique. Jevrejeva et al. (2014) [J2014] reconstructed the longest records for GMSL with the largest numbers of tide gauges using the station stacking method. Hay et al. (2015) [H2015] initiated the data assimilation approach by incorporating

622 tide gauges. The hybrid reconstruction by Dangendorf et al. (2019) [D2019] combined the data assimilation and EOF techniques to capture both long-term trends and interannual variability. Frederikse et al. (2020) [F2020] used the station stacking method to compute the GMSL for 1900-2018. Dangendorf et al. (2024) [D2024] developed a novel data assimilation approach that resolves spatial variability in sea level changes. Among these eight reconstructions, R2011 considered the smallest number of tide gauges. All these reconstructions span the twentieth century, with varying degrees of

coverage into the twenty-first century (see Table 2).

**Table 2. Sea level reconstruction from literatures.**

| Reconstruction | Data access | Reference | Method | Tide gauges | Time span |
|---|---|---|---|---|---|
| C2011 | https://www.cmar.csiro.au/sealevel/sl_data_cmar.html | Church & White (2011) | EOF | 642 | 1880-2013 |
| R2011 | https://psmsl.org/products/reconstructions/ | Ray & Douglas (2011) | EOF | 89 | 1900-2007 |
| J2014 | https://psmsl.org/products/reconstructions/ | Jevrejeva et al. (2014) | stacking | 1277 | 1807-2009 |
| H2015 | https://doi.org/10.1038/nature14093 | Hay et al. (2015) | assimilation | 622 | 1900-2010 |
| D2019 | https://doi.org/10.1038/s41558-019-0531-8 | Dangendorf et al. (2019) | hybrid | 622 | 1900-2013 |
| F2020 | https://doi.org/10.5281/zenodo.3862995 | Frederikse et al. (2020) | stacking | 559 | 1900-2018 |
| D2024 | https://doi.org/10.5281/zenodo.10621070 | Dangendorf et al. (2024) | assimilation | 516 | 1900-2021 |
| T2024 | https://doi.org/10.48364/ISIMIP.749905 | Treu et al. (2024) | hybrid | 622 | 1901-2015 |
| M2025 | https://doi.org/10.5281/zenodo.15385035 | This study | assimilation | 945 | 1900-2022 |

At local scale, our sea level reconstructions are compared to the reconstruction by Treu et al. (2024) [T2024]. Table 3 summarizes main features (pros and cons) in the reconstruction by T2024 compared to this study. The key feature of T2024 is that they synthesized sea level rise at local scale by integrating several different datasets. Their low-frequency relative sea

level changes (mainly reflecting sea level trends) are extracted from the combination of D2019 and Oelsmann et al. (2024). The former provides the geocentric sea level, and the latter releases the vertical land motion. The difference between those two variables defines the relative sea level, which is also reconstructed by this study. T2024 adopted an irregular grid covering global coast. If the sea level reconstruction by D2019 is not available at locations of this grid, T2024 interpolated or extrapolated the sea level based on D2019 grid. In addition, the reconstruction by D2019 employed projection from the

locations of tide gauges onto the satellite altimetry grid, which means D2019 does not directly cover the sites of tide gauges. We therefore must stress that the sea level reconstruction by T2024 is not built on the exact sites of tide gauges, which is a major difference from this study. An apparent advantage in T2024 is that they include high frequency variability in the reconstructed sea level time series. They released monthly reconstructions for 1901-1978, and hourly reconstructions for 1979-2015. However, our reconstructions do not contain high frequency sea level variability, which is a major limitation (see

section 4).



**Table 3. Comparisons of sea level reconstruction by T2024 and M2025.**

| Reconstruction source | M2025 | T2024 |
|---|---|---|
| Raw records | 945 tide gauges | 622 tide gauges |
| Reconstruction method | Data assimilation | Hybrid |
| Reconstructed trends (records available at a tide gauge) | Follow closely raw records | Differences are possible between reconstructions and raw records |
| Reconstructed trends (records NOT available at a tide gauge) | Physical interpolation and extrapolation | Mathematical adjustment by EOF reconstruction |
| Reconstructed variability | No high frequency variability | Monthly (1901-1978) and hourly (1979-2015) |
| Reconstruction field | The exact sites of tide gauges | A coastal grid, with spatial projection and interpolation |
| Reconstruction ensemble | 35 reconstructions | 1 reconstruction |

## 2.8 Satellite altimetry

We use monthly sea level time series provided by Archiving, Validation and Interpretation of Satellite Oceanographic (AVISO) service (https://www.aviso.altimetry.fr/en/home.html). This product is spatially gridded into a 0.25°×0.25° grid, which combines measurements from TOPEX/Poseidon, Jason-1/2/3, HY-2, Sentinel-3A, and Cryosat-2. Various geophysical corrections, e.g., wet troposphere correction, and atmospheric loading correction, have been applied to the AVISO grids. In addition to the gridded monthly products, AVISO also releases weekly GMSL time series that have been corrected for GIA effect. The time series are available at https://data.aviso.altimetry.fr/aviso-gateway/data/indicators/msl/. We average the weekly samples into annual time series, and compare both the AVISO gridded product and its GMSL time series to our sea level reconstructions.

## 2.9 Validation methodology

Our sea level reconstructions are validated through comparing with sea level observations and other sea level reconstructions. The validation process includes comparisons at global and local scales. At global scale, sea level reconstructions are commonly compared to the sea level rise observed by satellite altimetry, because it provides robust evidence for the GMSL rise. At selected locations, our reconstructions are compared to AVISO sea level observations. We use the AVISO time series to implement the comparison over 1993-2022, which means a limited period for comparison. To validate our sea level reconstructions over the twentieth century, we compare them with other sea level reconstructions. Several reconstructed GMSL time series are publicly available (Table 2), those sea level reconstructions are considered for the validation at global scale. Sea level reconstructions are valuable at local scale, but they are less available. T2024 made an effort to address this issue, and their sea level reconstruction offers an independent estimate for validating our sea level reconstruction. We average their monthly and hourly reconstructions into yearly time series, consistent with our reconstructions. However, as mentioned in section 2.7, T2024 reconstruction was performed on a coastal grid by interpolating or extrapolating the reconstruction by D2019, they do not directly provide time series at the exact sites of tide gauges. To implement the comparison, we select the nearest grid point from T2024 for each site of tide gauge considered in this study.



## 3 Results

In this section, we present the main results, beginning with several examples of sea level reconstruction at different tide gauges that illustrate diverse reconstructions at tide gauges. They are followed by comparisons between our reconstructions and other estimates, including observations from satellite altimetry and other sea level reconstructions. Those comparisons 315 serve to verify our reconstruction at global and local scales, and elaborate the merits and limitations in our reconstructions. The final subsection is committed to address the statistical assessment, e.g., spread, median, or a particular percentile.

### 3.1 Examples of sea level reconstruction

Figure 6 plots reconstructed sea level time series at four selected tide gauges. These examples highlight the presence of substantial data gaps over 1900–2022. For instance, Daugavgriva, station ID 37, ceased to record sea level since 1940 320 (Figure 6a). Dunkerque, station ID 468, started to observe sea level since around 1950 (Figure 6c), but it is also associated with a data gap from 1980 to 2000. Sokcho, station ID 1365, only covered a short time duration (Figure 6d). Our sea level reconstructions fill in those gaps, regardless their durations. More importantly, our reconstructions are physical interpolations/extrapolations, because they accommodate (physically) simulated sea level from climate models and (physically) predicted sea level owing to water exchange between land and oceans (i.e., the GRD effect).

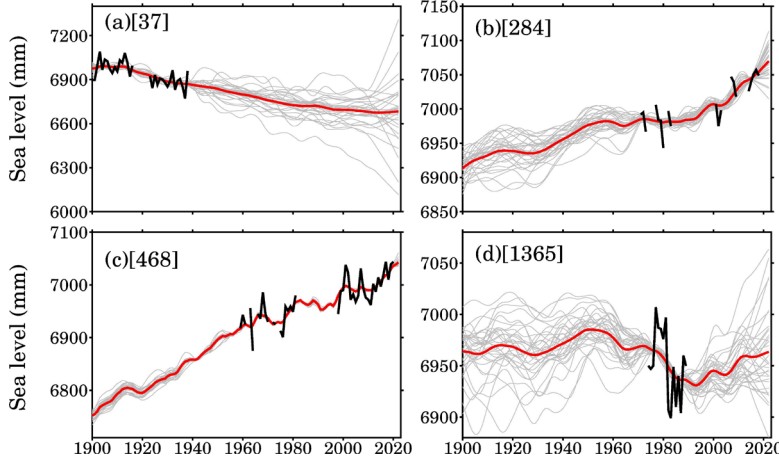


**Figure 6: Examples of sea level reconstructions at selected tide gauges. (a) Daugavgriva, PSMSL ID 37; (b) Durban, PSMSL ID 284; (c) Dunkerque, PSMSL ID 468; (d) Sokcho, PSMSL ID 1365. Black lines are raw records from PSMSL; gray lines are individual reconstructions derived with 35 climate models using our assimilation framework at the same locations of tide gauges; red lines represent the ensemble mean.**

Two notable features emerge from our sea level reconstructions. First, the 35 reconstructed sea level time series are characterized by smooth and refined curves with reduced year-to-year fluctuations compared to the raw tide gauge records (Figure 6). Second, the 35 reconstructed time series tend to converge over periods when raw records are available, underscoring the strong influence of sea level observations in constraining the reconstructions. In contrast, in the absence of



observations (i.e., during data gaps), the reconstructions exhibit a wider range of behaviours, reflecting the inherent spread

among climate model simulations at local scales. In some cases, such as Figure 6a, the spread in the reconstructed sea level at the final time step (year 2022) can approach 1 m. The average of all 35 reconstructed curves, i.e., the red lines shown in Figure 6, suggests smoother and more refined sea level changes at tide gauges. This feature is expected, because the average tends to reduce variations across the ensemble.

## 3.2 Comparison with other estimates

### 3.2.1 Comparison with satellite altimetry

We first compare our reconstructions with observations from satellite altimetry (Figure 7a). The average of the 35 GMSL reconstructions based on all 945 tide gauges yields a long-term trend of 3.52 mm/yr over 1993-2022, highly consistent with the trend (3.56 mm/yr) of the GMSL observed by satellite altimetry. However, this apparent consistency should be interpreted with caution, as it may involve differences in definitions of GMSL, inherent uncertainties, and

coincidental agreement. First, although both our sea level reconstructions and satellite observations are corrected for GIA effect, they represent fundamentally different quantities: our reconstruction reflects relative sea level at tide gauge locations, while satellite altimetry measures absolute sea level over the global oceans. Secondly, our reconstructions represent very limited samples of changes in sea level along coastal zone, it does not include any changes in the ocean interior, although it does cover high latitudes, where are not observed by satellite altimetry.

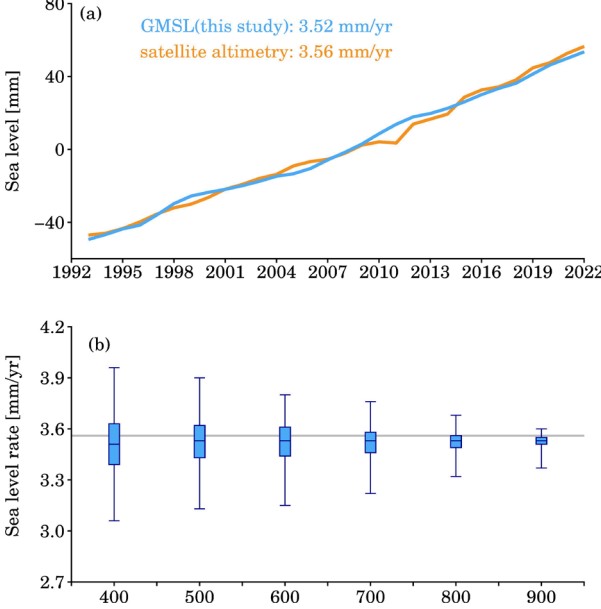


**Figure 7: Global mean sea level (GMSL) and its rate over 1993–2022. (a) Blue line is the average of 945 tide gauges, and orange line is provided by AVISO (section 2.8); (b) boxplots show ensemble mean rates using different numbers of tide gauges, gray line indicates the GMSL trend from AVISO.**

The samples (numbers and distributions) of tide gauges indeed affect the average sea level rates. To evaluate this effect,
we extract a subset from the reconstructed sea level at total tide gauges. The ensemble of subset ranges from 400 to 900 with
a 100 interval. For each subset, we randomly repeat the spatial resample for 1000 times, then compute the median (50th
percentile), maximum (100th percentile), minimum (0th percentile), and quartiles (25th percentile and 75th percentile). The
resulting statistics (Figure 7b) indicate that a small subset has a wide range between the maximum and minimum. For
example, for the subset of 400, the maximum rate is 3.96 mm/yr, and the minimum rate is 3.06 mm/yr, yielding a 0.9 mm/yr
total range. This range is reduced to 0.23 mm/yr for the subset of 900. We also note that the median is very close to the
GMSL rate from satellite altimetry, regardless the number of subsets.

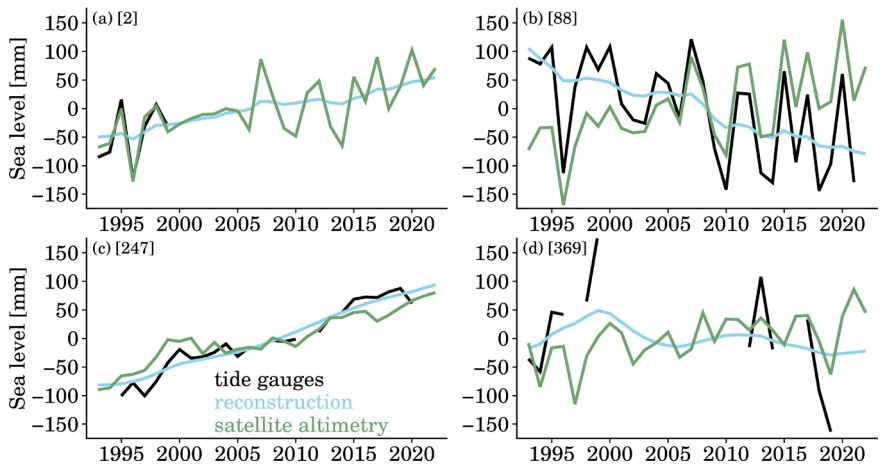

**Figure 8: Sea level time series at selected tide gauges. (a) Swinoujscie, PSMSL ID 2; (b) Ratan, PSMSL ID 88; (c) Port
Lyttelton, PSMSL ID 247; (d) Garden Reach, PSMSL ID 369. Black lines are raw records from PSMSL, green lines
are observed by satellite altimetry, and the blue lines are the average time series reconstructed by this study.**

Our sea level reconstructions are also compared against observations from satellite altimetry at some locations. Figure 8
shows the comparisons at four selected tide gauges. The blue lines in Figure 8 are the average of 35 sea level reconstructions.
Since satellite altimetry products usually do not provide direct estimates at the sites of tide gauges, we consider the nearest
grid point from AVISO grids within 50 km. At some locations (e.g., Figure 8a and 8b), tide gauges exhibit very similar
variability to the satellite altimetry, although their trends might be apparently different. The difference in sea level trends
could be attributed to several factors. For instance, a major reason is related to the local vertical land motion (Woppelmann
& Marcos, 2016), because tide gauges observe relative sea level changes, but the satellite altimetry monitors the absolute sea
level changes, variations in local vertical land motion would cause difference in those two observations. Other factors could
be related to errors in observations, and local forcing (e.g., Woodworth et al., 2019; Piecuch et al., 2019). We note that, at
some sites of tide gauges, our sea level reconstructions agree with the observations from satellite altimetry, even if the
records of tide gauges are not available (Figure 8a and 8d). However, this agreement is built on the average of
reconstructions, there could be larger discrepancies in individual reconstruction (for instance, see Figure 6).

### 3.2.2 Comparison with other sea level reconstructions

We compare our reconstructions to other sea level reconstructions for the GMSL (Figure 9). When determining the
GMSL rate, we consider the period 1900-2007, because this period is commonly covered by all reconstructions. Overall, our
reconstructed GMSL curve aligns with other reconstructed GMSL curves, representing a new, independent estimate of
GMSL rise. Those curves generate GMSL rates ranging from 1.31 mm/yr to 1.97 mm/yr. The highest rate is determined by
J2014 who employed a station stacking method. The lowest rate is identified by H2015 who proposed the data assimilation
approach. Our curve yields a rate of 1.60 mm/yr, very close to the rate of 1.62 mm/yr by C2011.

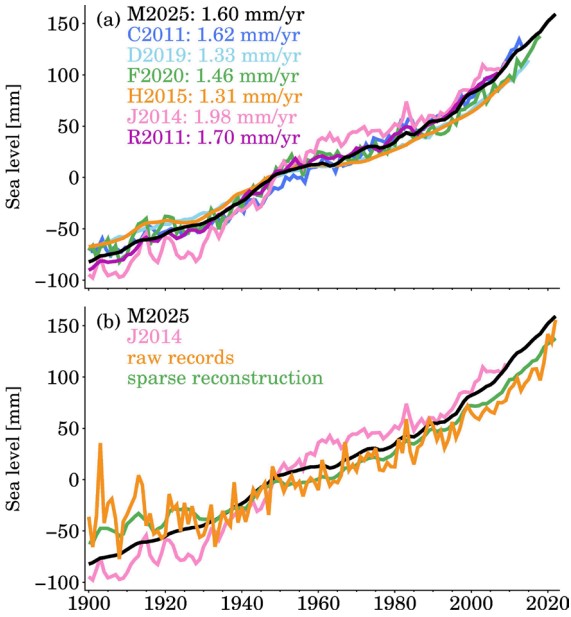


**Figure 9: Global mean sea level rise since 1900. (a) compares different sea level reconstructions from this study
[M2025], Church & White (2011) [C2011], Ray & Douglas (2011) [R2011], Jevrejeva et al. (2014) [J2014],
Dangendorf et al. (2019) [D2019], and Frederikse et al. (2020) [F2020], the rates are estimated for 1900-2007, as this
period is covered by all reconstructions. (b) 'raw records' is the GMSL that is recomputed with raw records from the
tide gauges selected by this paper, 'sparse reconstruction' is the GMSL that uses the reconstructed sea level when
raw records are available (i.e., with data gaps).**

The discrepancies among these reconstructions can be largely attributed to the methodology, but the selection and
distribution of tide gauges also plays an important role. For instance, while both C2011 and R2011 employed the classic
EOF reconstruction method, they yielded substantially different estimates of GMSL rise (Figure 9a). This discrepancy
primarily stems from the fact that R2011 incorporated only 89 tide gauges, whereas C2011 utilized more than 500 tide
gauges (Table 2). The spatial coverage of tide gauges strongly influences the resulting reconstruction, as also demonstrated
by our resampling experiment (Figure 9b) and Hamlington & Thompson (2015).



It is noteworthy that the reconstruction by J2014 shows the largest interannual variability (Figures 9). We suspect that these fluctuations are caused by direct average from raw records. To test this hypothesis, we construct two additional GMSL

time series: one based on the raw tide gauge records, and another based on our reconstruction, restricted to periods when raw records are available (Figure 9b). The resulting GMSL curve with raw records exhibits a large interannual variability, similar to the result of J2014, confirming our conjecture. Interestingly, Figure 9b also suggests a lower GMSL rise during the early twentieth century (1900–1930) compared to our full reconstruction and that of J2014. This discrepancy may be linked to data gaps (when comparing raw records with our reconstruction) and the smaller number of tide gauges used (when comparing

with J2014).

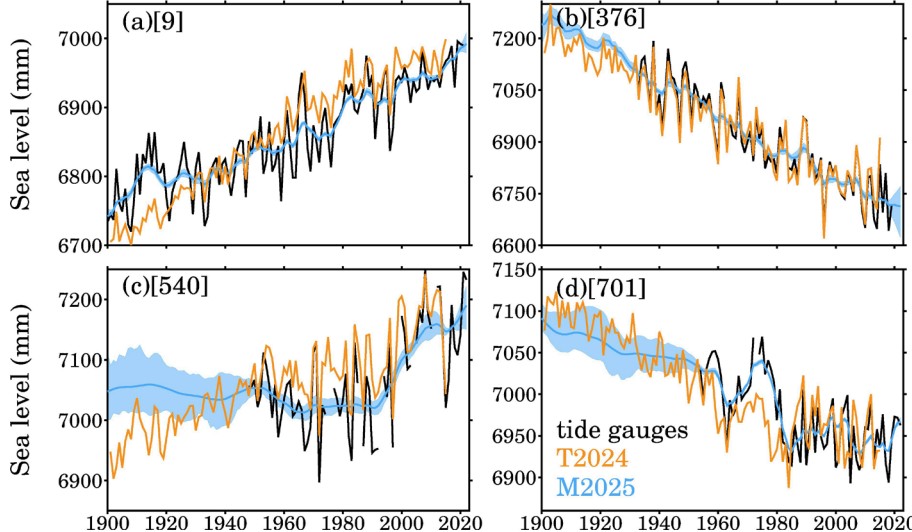

**Figure 10: Comparisons of sea level reconstruction at selected tide gauges. (a) Maassluis, PSMSL ID 9; (b) Rauma, PSMSL ID 376; (c) Apra Harbor, PSMSL ID 540; (d) Kainan, PSMSL ID 701. Black lines are raw records from PSMSL, orange lines are reconstructed by Treu et al. (2024) [T2024], blue lines are the medians reconstructed by this**

**study [M2025], and the light blue shading indicates the uncertainty bounded by 10th percentile and 90th percentile.**

Our sea level reconstructions at tide gauges are compared to the time series reconstructed by T2024. Figure 10 shows the comparisons at four selected tide gauges. It is clearly noted that the sea level reconstructions by T2024 are associated with high-frequency variations (year-to-year fluctuations), which are also suggested by the raw records, those fluctuations are even highly consistent at some tide gauges, e.g., ID 376 (Figure 10b). However, there are also apparent discrepancies in

low frequency changes, e.g., tide gauges ID 9 and ID 701. Those discrepancies probably originate from the covariance difference between tide gauges and satellite altimetry, as the latter's covariance is used to determine the sea level reconstruction. On the other hand, our sea level reconstructions closely align with the raw records, because in our data assimilation, those raw records are employed to constrain the sea level reconstructions, or in other words, our sea level reconstructions always follow the raw records, underscoring the importance of the observed evidence for sea level rise. From

Figure 10, we observe evident difference between the sea level reconstruction by T2024 and our sea level reconstructions,



especially when the raw tide gauges are not available, e.g., Figure 10c. In those situations, the sea level reconstructions are essentially extrapolated given the reconstruction methods or information, our extrapolations are mainly based on the sea level physics (model simulations and predictions, see section 2 and Table 3), while T2024 relies on the covariance from satellite altimetry.

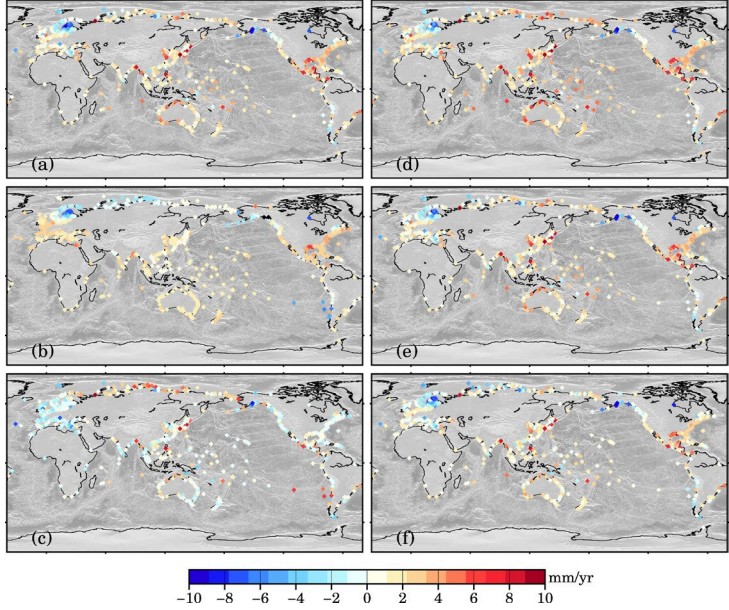


**Figure 11: Sea level rate at tide gauges for 1901-2015. (a) This study, average rate of 35 sea level reconstruction; (b) T2024; (c) difference between this study and T2024; (d) 95th percentile rate of 35 sea level reconstruction; (e) 50th percentile rate of 35 sea level reconstruction; (f) 5th percentile rate of 35 sea level reconstruction.**

Different reconstructions may indicate diverse sea level trends. We compute the sea level rate at all tide gauges for 1901-2015 (Figure 11), and evaluate the rate difference between our sea level reconstruction and the reconstruction by T2024. Note that our sea level rates are computed from the average of the ensemble of sea level reconstructions. The rate comparison suggests that our sea level reconstruction is associated with high spatial variability, for instance, along the Arctic coast, South America coast, and around the Pacific Oceans, on the other hand, the sea level reconstruction by T2024 is characterized with a rather smoothed pattern, especially around the Europe coast (excluding the Baltic Sea), Asia east coast and Australia coast. Indeed, the rate differences are evident along the Arctic coast, the South America coast. We identity a rate of 9.25 mm/yr for the maximum difference, and a rate of -7.78 mm/yr for the minimum difference. Despite the spatial difference, on average, our sea level rates are surprisingly consistent with the sea level rates by T2024, two datasets generate average rates of 1.22 mm/yr and 1.20 mm/yr, respectively, associated with standard deviations of 2.42 mm/yr and 2.08 mm/yr, respectively.

Another advantage in our sea level reconstructions is that, unlike the reconstruction by T2024 who provide only a single time series, we provide an ensemble of sea level reconstructions that include 35 time series at each tide gauge (see Figure 6).



We can compute either the average (red lines in Figure 6) or the median (blue lines in Figure 9) using those 35 complete time series, and use the average or median to represent the robust sea level reconstruction at tide gauges. We find that at most tide gauges, the average and the median are almost identical (Figure 11a and 11e). In addition, the ensemble of our sea

level reconstructions permits for the computation of a particular percentile, for example, the 90th and 10th percentiles, and those two percentiles can form boundaries for uncertainties (see the light blue shading in Figure 9). In the following subsection, we illustrate how to assess the sea level rate using the ensemble of our sea level reconstructions.

### 3.3 Statistical assessment

In this subsection, we present statistical assessment for sea level rise using our reconstructed time series. Given the

ensemble of 35 sea level reconstructions (see Figure 6 for examples), we can compute average, spread, median, or a particular percentile for both sea level rates and sea level curves. For instance, Figure 6c shows 35 sea level reconstructions at tide gauge Durban, PSMSL ID 284. For each sea level reconstruction (or curve), we can compute a linear rate (or acceleration) over a period of interest. Figure 12 plots 35 sea level rate over 1900-2022 at tide gauge Durban for all sea level reconstructions. Those 35 rates range from 0.70 mm/yr (minimum, or 0th percentile) to 1.38 mm/yr (maximum, or 100th

percentile), with a median rate of 0.95 mm/yr, which is very close to the average rate of 0.97 mm/yr, as shown in Figure 12. There are two ways to estimate the uncertainty for the rate. We can compute the spread (i.e., standard deviation) using those 35 rates, or alternatively, we can compute percentiles (e.g., 10th and 90th) to form boundaries for the rate uncertainty (see Figure 12). At most tide gauges, we report that the rate differences are very minor (< 0.1 mm/yr, see Figure 10) between the median and the average, although several tide gauges are identified to have high value of rate differences (> 5 mm/yr),

because they have large abrupt jumps (see section 4) that affects the sea level reconstructions.

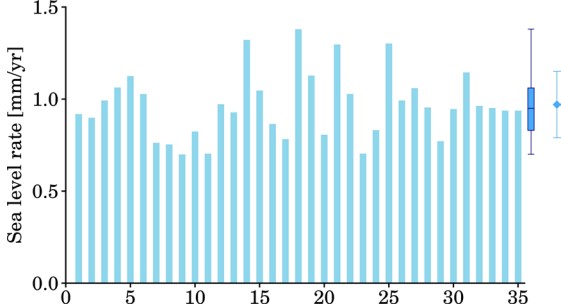

**Figure 12: Sea level rate over 1900-2022 at tide gauge Durban, PSMSL ID 284. The 35 thin rectangles represent sea level rates estimated from our 35 sea level reconstructions with climate models (see model index in Table 1), the 35 curves are shown in Figure 6b; the boxplot on the right side indicates the 0th, 25th, 50th (median), 75th, and 100th**

**percentiles, the diamond on the right side indicates the average rate and the error bar indicates the spread (i.e., standard deviation), the median rate is almost identical to the average rate.**

The rate spreads at tide gauges are demonstrated to be time-variable. We explore the rate spreads for two periods, 1900-2020 and 1900-1950 (Figure 13). Over these two periods, large spreads (> 0.8 mm/yr) are mainly shown along Arctic coast,





which are primarily attributed to the diversity in the SDSL changes. We also note that the spreads over 1900-1950 are larger than the spreads over 1900-2020, a major reason is that sea level rates are expected to be high over a short period, resulting in large spreads. Small spreads (< 0.4 mm/yr) are observed along the coast of India, North America, and Europe. Those small spreads are either caused by similar sea level reconstructions or small trends in sea level rise. In the former cases, raw records are available at most time points, leading to very similar reconstructions (see Figure 10a), as our sea level reconstructions follow raw records.

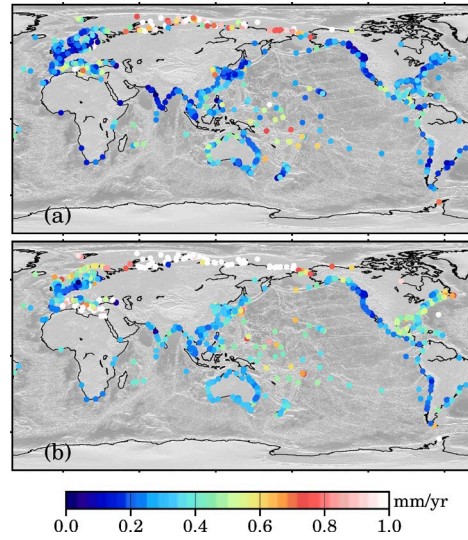


**Figure 13: Spread in sea level rate at tide gauges. (a) 1900-2020; (b) 1900-1950.**

**4 Caveats**

Our sea level reconstructions offer complete sea level time series, however, use of these data should be with cautions. They do not purely reflect sea level signals associated with the mechanisms defined in section 2.1, but also possibly contain

some changes due to other geophysical processes or anthropologic activities. It is well known that earthquakes cause abrupt jumps in the records of tide gauges (e.g., Oelsmann et al., 2024). For instance, the tide gauge Ofunato II (PSMSL ID 1364, located in Japan), recorded an abrupt uplift in sea level since 2011, amounting to about 680 mm (Figure 14). This sudden jump was clearly not caused by SDSL or SLF changes, but actually triggered by the Tohoku-Oki 2011 earthquake (Ozawa et al., 2011; Simons et al., 2011), which resulted in dramatic co-seismic displacement (downward) that consequentially

elevated the relative sea level. We also discover an evident decrease in sea level after that jump, which is also not purely relevant to SDSL or SLF changes, but mostly induced by post-seismic uplift, or viscoelastic relaxation (Han et al., 2019) that could persist for years or even decades. There are some similar cases at other tide gauges that experienced uplift or subsidence due to earthquakes. During the process of tide gauges selection, we did not remove those tide gauges associated with sudden jumps, or simply eliminate those abnormal records, because, first, we intend to include tide gauges as many as





possible, and second, those tide gauges or abnormal records have negligible effect on reconstructions at other tide gauges, although those abnormal records do affect our reconstruction at their tide gauges before and after the sudden jumps. Users should particularly pay attention to those jumps, and inspect the raw records before employing our reconstructions.

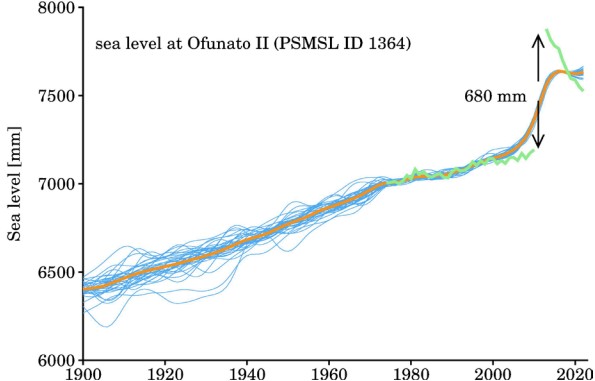

**Figure 14: Sea level at tide gauge Ofunato II, PSMSL ID 1364, Japan. Green line is the raw records, which were broken by the**
**2011 Tohoku-Oki earthquake (Ozawa et al., 2011). This earthquake caused a sudden jump of 680 mm sea level, followed by a rapid decline, which is probably not a sea level signal, but could be mostly attributed to Earth viscoelastic relaxation (e.g., Han et al., 2019). Blue lines are the ensemble of sea level reconstructions, and the orange line is their average. Since the jump is not removed, anomalous jumps are also manifest in our sea level reconstructions.**

Our sea level reconstructions are focused on the refined sea level trends, because the data assimilation approach is
informed with yearly rates only. The refined trends indeed benefit the study of low-frequency sea level rise, unfortunately, they do not reflect any year-to-year variability as shown in the reconstructions by T2024 (see Figure 10), and they are probably not suitable for determining trends over a short period. Although the long-term trend and short-term trend may not have an exact diacritical point, we recommend that sea level rates should be estimated over a period larger than 30 years (e.g., Frederikse et al., 2020; Wang et al., 2024). Sea level rates spanning period less than 30 years must be explained with
cautions, and their uncertainty might be greatly larger than the spread advised by our sea level reconstructions.

## 5 Conclusions

In this paper, we reconstructed sea level rise at global 945 tide gauges for 1900-2022 with a data assimilation approach (Hay et al., 2013; Mu et al., 2024a). This approach accommodates sea level simulations from climate models and sea level predictions with the GRD effects (Frederikse et al., 2016), therefore, the resulting sea level reconstructions are physical
interpolations and extrapolations. More importantly, by incorporating outputs from 35 climate models, the sea level reconstructions offer an ensemble of complete, refined, and smooth time series at tide gauges, and allow for direct uncertainty assessments that reflect reconstruction diversity or probability.

Global comparisons suggest that our sea level reconstructions align with sea level observations and other sea level reconstructions, demonstrating that our sea level reconstructions contribute to the ensemble of reconstructed GMSL curves



that are available to the community. In addition to exploring GMSL rise and acceleration, our GMSL time series can serve to validate other reconstructions, and estimate uncertainties. Local comparison with an independent reconstruction by T2024 indicates that our sea level reconstructions advocate the raw records of tide gauges, signifying that our reconstructions emphasize the importance of the observed evidence. Despite some trend differences from the reconstruction by T2024, our reconstructions are expected to contribute to the understanding of global sea level rise and its interplay with climate change.

**Code and data availability**

The released data 'SLRv2.nc' from our assimilation framework in this study can be accessed at:

https://doi.org/10.5281/zenodo.15385035 (Mu, 2025). It contains following variables:

-ID: it is a variable with dimension 945×1, which contains the ID assigned by PSMSL.

-lon: it is a variable with dimension 945×1, which contains the longitude of each tide gauge.

-lat: it is a variable with dimension 945×1, which contains the latitude of each tide gauge.

-year: it is a variable with dimension 123×1, the year from 1900 to 2022.

-sea_level: it is a variable with dimension 945×123×35, which contains the sea level reconstructions at all tide gauges over 1900-2022 for all 35 CMIP6 models.

-RSL: it is a variable with dimension 945×1, which contains the GIA relative sea level rates at all tide gauges.

-raw_records: it is a variable with dimension 945×123, which contains the annual records from PSMSL. Note that the missing values are assigned with 'NaN'.

-average: it is a variable with dimension 945×123, which contains the average of sea level reconstructions at all tide gauges over 1900-2022.

-spread: it is a variable with dimension 945×123, which contains the spread of sea level reconstructions at all tide

gauges over 1900-2022.

-GMSL: it is variable with dimension 123×1, which contains the global average time series of our sea level reconstructions at 945 tide gauges. Note that GIA RSL effect is removed.

-GMSL_spread: it is variable with dimension 123×1, which contains the spread of GMSL.

Our data assimilation was run with an open software *SSpace* (Villegas & Pedregal, 2018), which can be downloaded from:

https://doi.org/10.18637/jss.v087.i05. The scripts are also available upon request to mdp321@126.com.

**Author contributions**

DM conceived the idea, and performed the sea level reconstruction. DM and RH wrote the manuscript. PY, HY and TX discussed the results, edited and commented on the manuscript. TX acquired the fundings.



**Competing interests**

All authors have declared that none of the authors has any competing interests.

**Acknowledgements**

We thank the editor for valuable suggestions and comments that significantly improved the presentation of this work and the format of our dataset. We thank the authors who shared their sea level reconstruction with community. This research was funded by the National Natural Science Foundation of China (Grant Nos. 42192534 and 41904081). Ruhui Huang also
acknowledges funding from the PhD Fellowship of the State Key Laboratory of Marine Environmental Science at Xiamen University and the China Scholarship Council (202106310016).

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
