# Peer review of "Reconstructing sea level rise from global 945 tide gauges since 1900"

_Earth System Science Data, 2025_

## Referee Comment (RC2)

Dear Editor,

Thank you for the opportunity to review the manuscript "Reconstructing sea level rise at global 945 tide gauges since 1900" by Mu et al. This study introduces a new dataset of reconstructed sea level time series at 945 global tide gauge sites covering the period 1900–2022. The authors employ a data assimilation approach that integrates outputs from 35 CMIP6 climate models, sea level fingerprints (SLF), glacial isostatic adjustment (GIA) corrections, and a random process to capture unresolved local variability. Each tide gauge location is associated with a 35-member ensemble, allowing for physical interpolation across data gaps and direct quantification of reconstruction uncertainty. The results are evaluated against previous global mean sea level (GMSL) reconstructions and compared locally with an independent product by Treu et al. (2024). Overall, the dataset aims to improve the spatial and temporal completeness of historical sea level records while preserving physical consistency and enabling robust statistical assessments.

This manuscript presents an ambitious and valuable contribution by reconstructing a long-term sea level dataset directly at tide gauge locations, using an ensemble-based data assimilation framework. It offers methodological advances by extending previous assimilation techniques, resolving sea level changes explicitly at gauge sites rather than interpolated grids, and enabling uncertainty quantification through a 35-member ensemble. However, the scientific motivation behind reconstructing sea level specifically at tide gauges—as opposed to existing gridded products—requires clearer articulation. While the technical execution is sound, the manuscript would benefit from improved clarity in its writing and structure, as well as a more critical discussion of key assumptions, particularly the use of coarse-resolution climate model outputs to inform local-scale variability.

1. Motivation of the Work

While the authors present a technically sound reconstruction effort, the manuscript lacks a compelling justification for why this new reconstruction is necessary—particularly at the exact locations of tide gauges. Existing products already provide gridded or interpolated sea level fields that span the 20th century, and the advantages of reconstructing sea level directly at the gauge sites, rather than relying on interpolation from existing reconstructions, are not fully explained. It remains unclear whether the primary purpose is to improve regional and coastal estimates, fill data gaps, validate climate models, or support impact studies. Furthermore, the distinctions between this dataset and other recent efforts, such as Treu et al. (2024), Dangendorf et al. (2024), or Frederikse et al. (2020), are only briefly addressed in a comparison table, without a deeper discussion of functional or practical differences. A clearer articulation of the scientific and applied motivation would significantly strengthen the manuscript.

2. Methodology

The central methodology relies heavily on outputs from CMIP6 climate models to estimate sterodynamic sea level (SDSL) changes, which are used to fill data gaps at the tide gauge sites. However, the coarse spatial resolution and limited representation of shelf dynamics, coastal processes, and tectonic settings in global climate models are not sufficiently acknowledged.

While the authors introduce a random term to account for local variability, it is unclear whether this compensates adequately for biases or structural mismatches between models and observations at local scales. The manuscript would benefit from a more explicit discussion of the limitations of applying global climate model output to local-scale reconstruction, and from a clearer justification of the confidence placed in these physically driven interpolations at individual tide gauges.

**3. Validation**

Although the authors validate their reconstructions at the global scale by comparing with satellite altimetry and other GMSL products, the evaluation at local scales remains limited. In particular, more rigorous assessments are needed in regions affected by vertical land motion, tectonics, or discontinuous observational records. While qualitative comparisons at selected sites are shown, these do not fully demonstrate the fidelity of the reconstructions. To improve confidence in the dataset, the authors should present additional quantitative validation—such as RMSE, correlation, or explained variance—between the reconstructed and raw records at long, continuous tide gauge sites. Ideally, the analysis would also identify regions where reconstructions are more or less reliable, based on observational completeness or environmental complexity.

**4. Interpretation of Ensemble Spread**

The use of a 35-member ensemble to express uncertainty is a valuable feature of the reconstruction, but the interpretation of this spread is not sufficiently clear. The ensemble is constructed from 35 climate model realizations, which likely reflect structural differences in the models and their simulation of large-scale processes. However, this ensemble does not appear to incorporate observational error, methodological uncertainty (e.g., parameter tuning), or other sources of reconstruction variability. Presenting the ensemble spread as a comprehensive uncertainty estimate may therefore be misleading. The authors should clarify what the ensemble spread represents—and, just as importantly, what it does not—and consider discussing additional sources of uncertainty that are not captured by this approach.

**5. Data Usability**

The caveats section correctly notes that some tide gauge records include abrupt jumps due to earthquakes or other geophysical events, which are then inherited by the reconstructions. However, the manuscript does not offer a systematic way for users to identify or handle these problematic records. For a dataset intended to support broad scientific and applied use, this raises concerns about usability and transparency. At minimum, the authors should consider flagging affected sites or events within the data files, and ideally provide guidance on how users might treat such anomalies (e.g., masking, correction, or exclusion). More generally, the caveats section would be more helpful if integrated earlier in the manuscript and more clearly linked to the limitations of the reconstruction method.

Minor Comments:
Title: Consider rewording for clarity, e.g., "Reconstructing global sea level rise from 945 tide gauges since 1900" is smoother.

Line 7: "Tide gauges record sea level changes along coast." → "along the coasts" or "along coastlines"

Line 10: "sometime persistent" → should be "sometimes persistent"

Line 15: "offering complete and refined sea level time series" → "providing continuous and refined sea level time series" might read better.

Line 18: "agreements" → "agreement"

Line 19: "despite apparent rate differences at locations, it is suggested…" → This phrasing is awkward. Suggest: "Despite some rate differences at certain locations, the reconstructed trends closely follow the raw records…"

Line 22: "informing coastal adaptation strategies" – consider specifying how this is useful, even briefly.

Line 27: "Tide gauges sample relative sea level changes along coast." → should be "along the coast" or "along coasts"

Line 40: "characterized with" → should be "characterized by"

Line 41: "only, (see Figure 1b)" → comma before parenthesis is awkward; rephrase as "e.g., only a few years (see Figure 1b)."

Line 47: "as well as spatial and temporal interpolation or extrapolation using neural networks…" – awkward phrasing. Suggest breaking into two sentences or removing "as well as".

Line 59: "added it into the basic functions" → "added it to the basic functions"

Line 64: "some major climate variability such like the El Niño–Southern Oscillation" → "such as"

Line 79: "the neural networks" → "neural networks"

Line 89: "extrapolations on rates" → better: "extrapolations of rates"

Line 93: "examination for reginal sea level rise" → should be "regional"

Line 100: "distinguished literatures" → "seminal studies" or "notable publications"

Line 104: "use to reconstruct" → "use it to reconstruct"

2.1 Title: "Sea level reconstruction by data assimilation" → Consider: "Sea level reconstruction using data assimilation"

Line 110: "to facilitate understanding for readers" → redundant; delete or simplify: "to facilitate understanding"

Line 116: "physically orientated" → should be "physically oriented"

Line 120: "including Greenland ice melting…" → better as "including mass loss from the Greenland Ice Sheet…"

Line 124: "Those oceanic geometries are termed as sea level fingerprint" → "These oceanic patterns are termed sea level fingerprints"

Line 126: "A random process is further proposed…" → awkward. Try: "We also introduce a random process…"

Line 200: "clime model" → "climate model"

Line 215: "we do not exclude records with large jumps or high rates, as their impact… is negligible" → requires justification or citation.

2.6 GIA description: "mainly is an ongoing response…" → should be "is mainly an ongoing response…"

Line 254: "see section 'Code and data availability'" → inconsistent with other section references; consider standardizing.

2.7: "Reconstruction from literatures" → should be "Reconstructions from previous studies" or "Existing reconstructions"

Line 265: "exerted broad influence" → more objective phrasing is "widely used"

Table 2 Caption: "Sea level reconstruction from literatures" → "Overview of sea level reconstruction studies"

2.8: "we average the weekly samples into annual time series…" → passive form might be clearer: "The weekly data were averaged to annual time series…"

2.9: "we select the nearest grid point from T2024 for each site of tide gauge" → "…for each tide gauge site"

Line 315: "committed to address" → "dedicated to addressing"

Line 318: "illustrate diverse reconstructions at tide gauges" → redundant phrasing. Better: "illustrate the diversity in reconstructions"

Line 323: "regardless their durations" → "regardless of their duration"

Line 324–326: Repetition of "physically" in "(physically) simulated sea level…" is awkward and unnecessary.

Line 332: "tend to converge over periods when raw records are available" → could be shortened: "converge when raw records are available"

Line 460: "rate differences are very minor" → better: "rate differences are generally small"

Line 470: "sea level rates are expected to be high over a short period" → maybe clarify: "rate estimates are more variable over short periods"

Line 477: "use of these data should be with cautions" → "should be used with caution"

Line 480: "anthropologic activities" → "anthropogenic activities"

Line 485: "not purely relevant to SDSL or SLF changes" → unclear. Better: "not directly attributable to SDSL or SLF mechanisms"

Line 490: "we did not remove those tide gauges… because, first, we intend to include…" → awkward. Suggest breaking into two sentences and rewriting as:

"We retained all gauges to maximize spatial coverage. Moreover, the impact of anomalous records is localized and does not significantly affect other stations."

Line 504: "Sea level rates spanning period less than 30 years must be explained with cautions…" → "Sea level rates estimated over periods shorter than 30 years should be interpreted cautiously…"

Line 510: "offer an ensemble of complete, refined, and smooth time series" → could be shortened: "provide refined, continuous time series"

Line 514: "align with sea level observations and other sea level reconstructions…" → redundant use of "sea level"; remove one.

Line 517: "our reconstructions advocate the raw records of tide gauges" → "closely follow" or "are consistent with" is clearer than "advocate"

Line 519: "expected to contribute…" → "expected to support efforts to understand…"

Line 521: "It contains following variables" → "It contains the following variables:"

Line 530: "missing values are assigned with 'NaN'." → "missing values are denoted by 'NaN'."

Line 534: "contains the spread of sea level reconstructions…" → maybe clarify: "the ensemble spread (standard deviation) across models"

Line 539: "scripts are also available upon request to…" → better to specify whether code will be publicly released or must be requested; ESSD encourages transparency.

---

## Author Comment (AC1)

The paper reconstructed a century-scale sea level rise at tide gauges, using a data assimilation approach that has been proposed by previous studies, but I saw some modifications or improvements, e.g., introducing a random process. The data assimilation is indeed driven by physical mechanisms, and therefore the reconstruction, or essentially the interpolation or extrapolation are physically interpretable. Authors considered 35 CMIP6 model output and, consequently, they gave 35 reconstructions, this ensemble obviously offers some useful statistical assessments, and this is really convenient, users can compute a desirable uncertainty estimate. Authors compared their reconstructions to observations from satellite altimetry, and other sea level reconstructions that are widely accepted by the community. The comparisons were performed on both global and local scales, and the results seems promising, although some differences were still seen, especially at the sites of tide gauges. The new global mean sea level reconstruction can serve as an independent estimate, users can get a better ensemble for average and spread. Overall, I think the dataset can be potentially applied to sea level studies, and the community would benefit from it. However, I have several comments, and I hope authors can address them before I see the paper published.

Reply: thank you very much for these comments that summarize concisely our work, and we sincerely thank you for your suggestions that help to improve the paper. Below, we answer your questions and address your concerns.

(1) In the method section, authors used the HP filter to compute the instantaneous rate for SDSL changes. My question is how the authors determine the parameter lambda (i.e., equation 14)? As a filter, HP might be sensitive to the changes of lambda. Based on my own understanding, the filtered or smoothed SDSL is perhaps related to the smoothed sea level curves seen in, e.g., Figure 6. Another reason that might explain the smoothed curves is that authors used the Kalman filter and smoother for the sea level fingerprints, so the total sea level would be much smooth. Authors need to prove how the curves would vary with parameter lambda.

Reply: this is good question. It should be recognized that the time series smoothed by HP filter are indeed affected by the parameter lambda. To illustrate this effect, we select the SDSL time series from the ACCESS-CM2 model interpolated at tide gauges Den Helder (PSMSL ID 23) and Buenos Aires (PSMSL ID 157), then perform the HP filter with lambda = 1, 10, 100, and 1000, respectively. See the plot below.

The plot shows the HP filtered time series at (a) Den Helder (PSMSL ID 23) and (b) Buenos Aires (PSMSL ID 157), with different values of lambda, note that the gray lines are the raw SDSL time series from the ACCESS-CM2 model. We can observe that a large lambda produces a refined curve that suffers from less high-frequency variability or better represents the low-frequency variability. In our practice, we adopt the value of 10, as it already shows less peak-to-peak changes. We must admit that this choice is empirically determined. This smooth curve is then used to

compute the instantaneous rates that drives the data assimilation, and yes, it is a major reason that our reconstructed sea level curves are smooth. The other reason is the application of Kalman filter and smoother, as you pointed out. All these materials are included in the revision, thank you again.

(2) There might be confusing explanation in Table 3. Treu et al. (2024) used the low-frequency sea level reconstruction from Dangendorf et al. (2019), who employed a hybrid reconstruction. This hybrid reconstruction combined traditional EOF reconstruction and the data assimilation, the former provided sea level variability, the latter provided long-term trends. But why authors claimed that 'Differences are possible between reconstructions and raw records', is this because they observed apparent discrepancies in Figure 10 when they compared with Treu et al. (2024). If so, I think there might be another reason, that is Treu et al. (2024) considered different selection of tide gauges. Anyway, authors should add some more wording to clarify.

Reply: thank you for the concern. We clarify this difference with more words. Please let us explain a little bit here. First, the selection of tide gauges has direct effect on the sea level reconstruction, this is no doubt. Second, we need to explain how the EOF reconstruction works. An important data processing procedure is that the sites of tide gauges should be projected onto the altimetry grid, which is commonly regular, e.g.,  $1^{\circ} \times 1^{\circ}$ , or  $0.25^{\circ} \times 0.25^{\circ}$ . To this purpose, we can search for the nearest altimetry grid point. Each site only has one nearest altimetry grid point, but, be careful, an altimetry grid point may be accompanied with two or even more sites. In such cases, we can merge these sites into one synthesized series of observations at the grid point. Anyway, the EOF reconstruction provides reconstructed sea levels on this regular grid, not at specific sites of tide gauges. So, even at a tide gauge that selected by both our study and Treu et al. (2024), there may be some differences.

(3) In Figure 9, authors compared many GMSL reconstructions to justify theirs. I saw some differences in the overall rates. Authors attributed the differences to reconstruction methods and selections of tide gauges, this is true, and I agree. But authors overlooked another fact, that is, the GMSL curves represent the relative sea level or absolute sea level? This is of course highly related to the reconstruction methods, but I think author should add some comments to this point, and the paper Dangendorf et al. (2017) might be helpful (https://doi.org/10.1073/pnas.1616007114). Reply: Thank you very much for this concern, we reflected on it, and thank you for providing the nice paper by Dangendorf et al. (2017). There is a subtle difference among the reconstructed GMSL time series; some of them are absolute GMSL and others are relative GMSL; the paper by Dangendorf et al. (2017) helps us to clarify. We are sorry that we ignored this issue before. Now we look into it.

Essentially, the reconstructed sea level represents either relative sea level or absolute sea level, and it depends on the correction to tide gauges. There are two ways. First, some studies included the vertical land motion effect at the sites of tide gauges. By nature, tide gauges (relative sea level) + vertical land motion gives us the absolute sea level, which is consistent with the observations from satellite altimetry; so, it is preferable to do this way for EOF reconstruction. The vertical land motion is observed by, e.g., GNSS and InSAR, but those observations only span recent years or one or two decades, they are not available for long-term reconstruction. Given this, many studies consider the vertical land motion caused by GIA process, which only represent a portion of total vertical land motion. This is a limitation, but it is the best we can do for now.

Our strategy is to correct the tide gauges for relative sea level effect caused by GIA process. By doing this, we obtain contemporary relative sea level rise at the sites of tide gauges. Consequently, our reconstructed sea level is different from, e.g., C2011, R2011, and J2014; they all represent absolute sea level. We also highlight that the real difference between our reconstruction and C2011 or R2011 or J2014 can be illustrated by the following relation:

C2011 = total absolute sea level = tide gauges + vertical land motion (assume that vertical land motion is only caused by GIA) = our reconstruction + GIA relative sea level + vertical land motion.

If we also assume that vertical land motion is only related to GIA, then, C2011 = our reconstruction + GIA absolute sea level (or geoid). Note that the GIA geoid changes are smooth over oceans, compared to the relative sea level

and vertical land motion, see Figure 1 in Tamisiea (2011; <a href="https://doi.org/10.1111/j.1365-246X.2011.05116.x">https://doi.org/10.1111/j.1365-246X.2011.05116.x</a>). But we should be aware of a fact that, at local scale, vertical land motion is related to many processes, more than just GIA. On average, we should expect a smaller change in vertical land motion. Nevertheless, the difference in sea level reconstruction could be related to this issue.

**All the discussion above is included in the revision.**

(4) In section 4, authors pointed out some limitations in their reconstructions, this is very important and useful. Authors claimed that some abrupt changes are not removed from the raw records, and those changes are induced by earthquakes, but have no significant effect on sea level reconstruction, this is understandable, as they account for only a small portion. I could agree to this. However, authors should elaborate a little bit more on the second limitation. The refined trends could be informative for long-term changes, despite we did not know how 'long' it is. The community usually employs a 30-year window for computing GMSL rise, so why not compare all the GMSL curves for the 30-year-rate curves, I would expect some differences, even significant ones, readers can glean some useful knowledge, and it would be better if authors add more wordings.

Reply: thank you for this good suggestion. We compute the curves of 30-year sea level rates, starting from 1915, and ending at 1993, as the reconstruction R2011 spans the period 1900-2007, see the plot below.

Our rate curve falls between those curves, except for the beginning period (1915-1928) and the ending period (1980-1993), over these two periods, our rate curve lies at the upper bound. The rate curve of J2014 is apparently distinct from other curves, especially since 1930, this distinction is of course directly connected with the fact that J2014 employed the largest number of tide gauges, which is about twice or even triple the numbers of tide gauges used by others.

The selection of tide gauges is indeed an important factor that affects reconstruction. This can be further confirmed by the difference between C2011 and R2011, especially over 1950-1980; both studies employed the EOF reconstruction, but they used very different distributions of tide gauges, R2011 used only 89 tide gauges, the lowest number for sea level reconstruction, to our knowledge. A major reason for this lowest number is that R2011 resolved the datum issue in PSMSL tide gauges. Every tide gauge requires a datum, but it's unknown. R2011 modified the EOF reconstruction, so the approach permits for estimating the datums. We suspect that, to better resolve the datum, a smaller number of tide gauges is desirable.

We notice that the reconstruction methods also matter, which is demonstrated by the difference between C2015 and D2019, as they considered very similar distribution of tide gauges, D2019 adopted the trends from C2015, but D2019 reconstructed interannual variability with the EOF reconstruction. This difference also implies that the interannual variability has some noticeable effect on the 30-year rates. A very similar comparison is suggested by Wang et al. (2024; <a href="https://doi.org/10.1175/JCLI-D-23-0410.1">https://doi.org/10.1175/JCLI-D-23-0410.1</a>, see their Figure 6), This paper is cited by our work. All the discussion above is included in the revision.

The figure shows the curves of 30-year running rates from different sea level reconstructions. This plotting is

**included in the revision.**

Minor suggestions:

Line 13, period over 1900 to 2022 -> period from 1900 to 2022

Reply: thank you, we change it to 'from'

Line 17, assessment -> assessments

Reply: we change it, thank you.

Line 18, GMSL rise -> GMSL rate

Reply: we change it to 'rate'

Line 22, observed sea level rise at -> observations from

Reply: we change the wordings, thank you.

Line 27, collection -> collected Reply: we change it to 'collected'

Line 27, add the reference Holgate et al. (2013) after the website

Reply: we add this paper citation.

Lines 33 and 35, reference Calafat et al. (2022) was not shown in the reference list, correct it

Reply: it should be Calafat et al. (2022a) or Calafat et al. (2022b) and these two papers are included in the reference

list, thank you.

Line 43, cause ->causes

Reply: we corrected it.

Line 62, at tide gauges ->at the sites of tide gauges

Reply: we add the words. Thank you.

Line 69, physical -> physically

Reply: we corrected it. Thank you.

Line 75, error in citation of Calafat et al. (2022) as shown before

Reply: we corrected it. Line 76, Since -> since

Reply: we corrected it. Thank you.

Lint 97, at tide gauges -> at the sites of tide gauges

Reply: we corrected it.

Line 113, raw tide gauge records -> raw records of tide gauges

Reply: we corrected it. Thank you.

Line 132, include -> includes

Reply: we corrected it.

Line 176, its -> their

Reply: we corrected it.

Line 189, Instantaneous -> instantaneous

Reply: we corrected it. Thank you.

Line 190, variation -> variations

Reply: we corrected it.

Line 215, sea-level -> sea level, you should be consistent about the writing

Reply: we use consistent wordings. Thank you.

Line 226, to address the 'a zero global mean', you add ocean mass increase, what about the global mean of thermosteric sea level, how you exactly treat this term?

Reply: the CMIP6 models used by this study provide estimates of global mean thermosteric sea level, in addition to

the gridded products, so we can remove the global mean of the gridded products, and add the global mean thermosteric sea level back to the gridded products.

Line 230, figure 3 was not even cited, you might want to add some more wordings to describe the changes shown in figure 3

Reply: good idea, we fix the citation, and add more wordings.

Line 235, error in caption of figure 4, e.g., panel (e) was missing

Reply: we corrected it. Thank you.

Line 240, rises -> rise

Reply: we corrected the typo. Line 249, remove 'for this'

Reply: we removed it. Thank you.

Line 300, AVISO sea level observations -> AVISO sea level products

Reply: we corrected it.

Line 316, assessment -> assessments

Reply: we corrected it.

Line 333, influence of sea level observations in -> influence of sea level observations on

Reply: we corrected it. Thank you.

Line 355, ensemble of subset -> subsets

Reply: we corrected it.

Line 379, the GMSL -> the GMSL curves

Reply: we corrected it.

Line 401, The resulting GMSL curve with raw records exhibits -> The resulting GMSL curves with raw records exhibit

Reply: we corrected it. Thank you.

Line 425, figure 11, add a panel showing the difference between 95th and 5th percentiles

Reply: Good suggestion, thank you.

Line 429, sea level rate -> sea level rates

Reply: we corrected it.

Line 440, T2024 who provide -> T2024 who provided or provides

Reply: we corrected it.

Line 458, 3.3 Statistical assessment -> 3.3 Statistical assessments, this correction applies to others, e.g., line 459

Reply: we corrected it. Thank you.

Line 505, add a plot showing the 30-year rates for all GMSL reconstructions

Reply: Great suggestion; we add this plot, along with a few wordings.

---

## Author Comment (AC2)

**Dear Editor,**

Thank you for the opportunity to review the manuscript "Reconstructing sea level rise at global 945 tide gauges since 1900" by Mu et al. This study introduces a new dataset of reconstructed sea level time series at 945 global tide gauge sites covering the period 19002022. The authors employ a data assimilation approach that integrates outputs from 35 CMIP6 climate models, sea level fingerprints (SLF), glacial isostatic adjustment (GIA) corrections, and a random process to capture unresolved local variability. Each tide gauge location is associated with a 35-member ensemble, allowing for physical interpolation across data gaps and direct quantification of reconstruction uncertainty. The results are evaluated against previous global mean sea level (GMSL) reconstructions and compared locally with an independent product by Treu et al. (2024). Overall, the dataset aims to improve the spatial and temporal completeness of historical sea level records while preserving physical consistency and enabling robust statistical assessments.

This manuscript presents an ambitious and valuable contribution by reconstructing a long-term sea level dataset directly at tide gauge locations, using an ensemble-based data assimilation framework. It offers methodological advances by extending previous assimilation techniques, resolving sea level changes explicitly at gauge sites rather than interpolated grids, and enabling uncertainty quantification through a 35-member ensemble. However, the scientific motivation behind reconstructing sea level specifically at tide gauges—as opposed to existing gridded products—requires clearer articulation. While the technical execution is sound, the manuscript would benefit from improved clarity in its writing and structure, as well as a more critical discussion of key assumptions, particularly the use of coarse-resolution climate model outputs to inform local-scale variability.

Reply: Thank you very much for these comments, they nicely summarize the key points of our work. The coarse resolution of CMIP6 models is indeed a limitation, we haven't addressed it adequately, although, in theory, some of the CMIP6 models have been resolved at a 'high' resolution. In the revision, we assess the climate model outputs against ocean reanalysis; the resulting assessments give us some new insights; we must admit that we made some improvement, but we didn't really overcome the limitation of climate models in describing local changes.

We sincerely thank you for these important suggestions.

**1. Motivation of the Work**

While the authors present a technically sound reconstruction effort, the manuscript lacks a compelling justification for why this new reconstruction is necessary—particularly at the exact locations of tide gauges. Existing products already provide gridded or interpolated sea level fields that span the 20th century, and the advantages of reconstructing sea level directly at the gauge sites, rather than relying on interpolation from existing reconstructions, are not fully explained. It remains unclear whether the primary purpose is to improve regional and coastal estimates, fill data gaps, validate climate models, or support impact studies.

Furthermore, the distinctions between this dataset and other recent efforts, such as Treu et al. (2024), Dangendorf et al. (2024), or Frederikse et al. (2020), are only briefly addressed in a comparison table, without a deeper discussion of functional or practical differences. A clearer articulation of the scientific and applied motivation would significantly strengthen the manuscript.

Reply: thank you very much for this suggestion. Our primary purpose is to improve the regional estimates with filling data gaps; about the validating the climate models, or supporting impact analysis, our data could be helpful, but it really depends on the community how to use the data.

In the revision, the discussion engages with the differences or similarities in these papers relevant to our study. These studies, Frederikse et al. (2020) [F2020], Dangendorf et al. (2024) [D2024], Treu et al. (2024) [T2024], our work [M2025], along with Dangendorf et al. (2019) [D2019] are closely connected. Their relation is explained with an illustration, shown below. These reconstructed datasets are suitable for various studies/applications on different spatial scales, as they have their own merits, and of course, limitations. We explicitly explain their features in the

main text, with the illustration shown below, so, we won't repeat the words here.

The illustration shows the relation among the studies of D2019, F2020, D2024, T2024, and M2025. A (applications) and B (scales) indicate their suitable investigations at various spatial scales, for example, A1 means the reconstruction can be used for the sea level rise, and B2 means it can be suitable for basin scale. Colour red (e.g., A2 or B3) means there are limitations or not mature. SLR = sea level rise; SLB = sea level budget; ESL = extreme sea level. This plotting is included in the revised paper.

**2. Methodology**

The central methodology relies heavily on outputs from CMIP6 climate models to estimate sterodynamic sea level (SDSL) changes, which are used to fill data gaps at the tide gauge sites. However, the coarse spatial resolution and limited representation of shelf dynamics, coastal processes, and tectonic settings in global climate models are not sufficiently acknowledged. While the authors introduce a random term to account for local variability, it is unclear whether this compensates adequately for biases or structural mismatches between models and observations at local scales. The manuscript would benefit from a more explicit discussion of the limitations of applying global climate model output to local-scale reconstruction, and from a clearer justification of the confidence placed in these physically driven interpolations at individual tide gauges.

Reply: this is a great question, we didn't give it enough thoughts before, and now we reflect on it.

To assess the improvement by the random process, we compare the SDSL to ocean reanalysis ORAS5. The comparison spans the period 1958-2014, as this period is covered by both CMIP6 and ORAS5. There are three types of SDSL, (1) the original SDSL from CMIP6; (2) the estimated SDSL by our data assimilation, i.e., the original CMIP6 SDSL plus the random process; (3) the SDSL from ORAS5. We compute the linear rate at tide gauges, then, compare sea level rate from (1), and from (2), to rate from (3). Note that we should remove the global mean of ORAS5, and then add the CMIP6 global mean for each individual model; this means we have 35 ORAS5 SDSL time series at each site of tide gauges. Specifically, we have 35 original SDSL, 35 estimated SDSL, and 35 ORAS5 SDSL. And, of course, we have their average.

We begin with the comparison for average, see the plotting shown below. Honestly, the correlation among the three types of SDSL rates are very low, -0.05 between CMIP6 and ORAS5, it is improved to be 0.14 between our estimated SDSL and ORAS5. These two correlations, one is 'no correlation', and the other is 'weak', should prove the useful help from the introducing the random process.

Figures shows the SDSL rates at tide gauges using (1) our estimated time series, (2) ORAS5 time series, and (3) the original CMIP6 time series. All time series are averaged using the ensemble of 35 time series. The rates are estimated for period 1958-2014.

We also note that, at many tide gauges, our estimated SDSL have very large rates, larger than ORAS5 and CMIP6. We suspect that both ORAS5 and CMIP6 (tend to) underestimate the sea level rise at tide gauges. To prove this conjecture, we compare two reconstructions to tide gauges, see the figure shown below. The first reconstruction is the average of our sea level reconstruction (by data assimilation), the second reconstruction is computed using the sea level fingerprints + ORAS5 SDSL + GIA (relative sea level). To get robust trend, we only consider tide gauges have valid records >40 years over 1958-2014, this gives us 350 tide gauges. We can see that, our reconstruction closely aligns with the tide gauges, their standard deviations are consistent (3.4 mm/yr VS 3.3 mm/yr); but the reconstruction with ORAS5 underestimates the sea level rise (with a standard deviation of 2.4 mm/yr).

Figure shows the comparison of sea level rate estimated from tide gauges, our reconstruction, and another reconstruction with ORAS5 (i.e., SLFs + ORAS5 SDSL + GIA). Tide gauges have a standard deviation of 3.4 mm/yr, our reconstruction has a standard deviation of 3.3 mm/yr, the reconstruction with ORAS5 has a standard deviation of 2.4 mm/yr.

There is more, if we compare the rates for individual CMIP6 model.

Despite the weak correlation using the average, some CMIP6 models show correlations with larger values. The strongest correlation (0.51) is produced by model NorESM2-MM (No. 33 shown in Table 1 in the main text), see the plotting shown right below. Seven models show correlations larger than 0.3, BCC-ESM1 (0.31), CanESM5-1 (0.43), CMCC-CM2-HR4 (0.34), CMCC-ESM2 (0.39), EC-Earth3-Veg (0.35), HadGEM3-GC31-LL (0.39), and NorESM2-MM. But we find that 21 CMIP6 models have correlations weaker than the average.

Thank you very much for again for raising the question.

**3. Validation**

Although the authors validate their reconstructions at the global scale by comparing with satellite altimetry and other GMSL products, the evaluation at local scales remains limited. In particular, more rigorous assessments are needed in regions affected by vertical land motion, tectonics, or discontinuous observational records. While qualitative comparisons at selected sites are shown, these do not fully demonstrate the fidelity of the reconstructions. To improve confidence in the dataset, the authors should present additional quantitative validation—such as RMSE, correlation, or explained variance—between the reconstructed and raw records at long, continuous tide gauge sites. Ideally, the analysis would also identify regions where reconstructions are more or less reliable, based on observational completeness or environmental complexity.

Reply: thank you so much for your concerns and these specific suggestions; there are indeed important, and also very challenging, we don't think we are able to address them all at this point of time.

Rigorous assessments are important. Some regions may suffer from apparent effect from vertical land motion, and tectonics (we must admit that tectonics is not really our expertise). In theory, vertical land motion would cause discrepancy in observations between tide gauges and satellite altimetry. But we don't think our reconstruction has the ability to assess the vertical land motion with the altimetry, or, if you don't mind, vice versa.

You mentioned other indicators, such like RMSE, correlation, or explained variance. They are very useful to quantify the variability. But unfortunately, our reconstructions produce the long-term trends, or a low-frequency variation. They are not really suitable for our reconstructions.

We certainly hope our data can motivate the community to explore regional sea level rise, we are expecting that their findings could support our data assessment, or rebut.

We are sorry that we cannot address these concerns adequately.

**4. Interpretation of Ensemble Spread**

The use of a 35-member ensemble to express uncertainty is a valuable feature of the reconstruction, but the interpretation of this spread is not sufficiently clear. The ensemble is constructed from 35 climate model realizations, which likely reflect structural differences in the models and their simulation of large-scale processes. However, this ensemble does not appear to incorporate observational error, methodological uncertainty (e.g., parameter tuning), or other sources of reconstruction variability. Presenting the ensemble spread as a comprehensive uncertainty estimate may therefore be misleading. The authors should clarify what the ensemble spread represents—and, just as importantly, what it does not—and consider discussing additional sources of uncertainty that are not captured by this approach.

Reply: this concern is important, thank you. The spread in 35 reconstruction is indeed constrained with limitations, as it only reflects the CMIP6 model diversity, or essentially, the reconstruction diversity. There are several additional error sources that we should include, if possible, or at least discuss them.

The first one, as you already pointed out, is the observational error or instrumental error. Church et al. (2024) adopted a consistent 4 mm error for all monthly records, but we should recognize that errors vary by sites, as they are managed/operated by different authorities/countries.

The second one is the uncertainty in GIA outputs (relative sea level). Studies have demonstrated that the specific

choice of two parameters, mantle viscosity and lithosphere thickness, affect the outputs of GIA models. However, at the moment, only very limited GIA outputs are available to us, e.g., ICE-6G\_C and ICE-6G\_D, they are almost identical at the tide gauges considered in this study. We didn't really have the resources or ability to assess this uncertainty.

You mentioned the uncertainty owing to parameter tuning in the data assimilation. The parameter tuning affects the reconstructed sea level. A key parameter is the trend variation for SLFs and random process. In the section of method, we clearly state that this parameter is set to be 1 mm/yr (equation 11). We test different values for SLF, 0.5 mm/yr and 2 mm/yr, see the figure shown below. We can see that the resulting reconstructions are very similar. We admit that the trend variation for random process has larger effect, 0.5 mm/yr would give us a small spread in the reconstructions, and it would be larger for 2 mm/yr. We empirically set the trend variations to be same for SLF and random process.

Figure shows the reconstructed sea level at tide gauge PSMSL ID 157, with trend variations of 0.5 mm/yr and 2.0 mm/yr. The results are very similar.

If users are interested in absolute sea level rise, then they have to consider the uncertainty in the vertical land motion, this could be a different story, because our reconstructed sea levels are relative sea levels by nature. Considering that measurements of vertical land motion is not available through the 20th century, quantifying this error is really challenging (e.g., Santamaria-Gomez et al., https://doi.org/10.1016/j.epsl.2017.05.038, 2017).

Anyway, we clarify that our spread would very likely underestimate the uncertainty.

**5. Data Usability**

The caveats section correctly notes that some tide gauge records include abrupt jumps due to earthquakes or other geophysical events, which are then inherited by the reconstructions. However, the manuscript does not offer a systematic way for users to identify or handle these problematic records. For a dataset intended to support broad scientific and applied use, this raises concerns about usability and transparency. At minimum, the authors should consider flagging affected sites or events within the data files, and ideally provide guidance on how users might treat such anomalies (e.g., masking, correction, or exclusion). More generally, the caveats section would be more helpful if integrated earlier in the manuscript and more clearly linked to the limitations of the reconstruction method. Reply: Thank you very much for this concern, this is a practical issue. There are two ways to identify anomalies in records of tide gauges.

The first one, which we believe is the most reliable approach, is to inspect the records with eyes; we must say that this approach need experiences, users should be familiar with the applications of tide gauges. This approach indeed costs several hours, but it is efficient.

The second way is also very simple, we can differentiate the time series, and obtain the year-to-year changes, or month-to-month changes if we use monthly time series. In an earlier study, Church et al. (2004; <a href="https://doi.org/10.1175/1520-0442(2004)017<2609:EOTRDO>2.0.CO;2">https://doi.org/10.1175/1520-0442(2004)017<2609:EOTRDO>2.0.CO;2</a>.) used monthly time series, and they excluded the differentiated records larger than 250 mm. We don't think there are consistent standards or threshold value, users can make their own thresholds, accordingly. Anyway, using the first approach, we identify 13 tide gauges associated with anomalous records. Their PSMSL IDs are specified in the revised manuscript section 4. Overall, their impact on the GMSL is minor, but the difference of 0.08 mm/yr rate is also detectable, this is because there are several tide gauges affected by the Tohoku-Oki 2011 earthquake, the jumps > 600 mm are really significant, especially if they occurred at more than just one tide gauges.

Figure shows the GMSL rate using the total 945 tide gauges, and 932 tide gauges (excluding the 13 anomalous tide gauges). This should answer to one of your minor comments shown below.

Furthermore, we state the caveats earlier in the comparison with T2024.

**Minor Comments:**

Title: Consider rewording for clarity, e.g., "Reconstructing global sea level rise from 945 tide gauges since 1900" is smoother.

Reply: nice suggestion, thank you.

Line 7: "Tide gauges record sea level changes along coast." → "along the coasts" or "along coastlines"

Reply: we corrected it, thank you.

Line 10: "sometime persistent" → should be "sometimes persistent"

Reply: we corrected it, thank you.

Line 15: "offering complete and refined sea level time series" → "providing continuous and refined sea level time series" might read better.

Reply: thank you for this suggestion.

Line 18: "agreements" → "agreement"

Reply: we corrected it, thank you.

Line 19: "despite apparent rate differences at locations, it is suggested..."  $\rightarrow$  This phrasing is awkward. Suggest: "Despite some rate differences at certain locations, the reconstructed trends closely follow the raw records..."

Reply: we corrected it, thank you.

Line 22: "informing coastal adaptation strategies" – consider specifying how this is useful, even briefly.

Reply: good suggestion, we specify briefly. Our time series can offer an insight into the sea level rise over the past century, especially if local records are not complete.

Line 27: "Tide gauges sample relative sea level changes along coast."  $\rightarrow$  should be "along the coast" or "along coasts"

Reply: thank you for the correction, we adopt it.

Line 40: "characterized with" → should be "characterized by"

Reply: thank you, we change the wording.

Line 41: "only, (see Figure 1b)"  $\rightarrow$  comma before parenthesis is awkward; rephrase as "e.g., only a few years (see Figure 1b)."

Reply: thank you, we change it.

Line 47: "as well as spatial and temporal interpolation or extrapolation using neural networks..." – awkward phrasing. Suggest breaking into two sentences or removing "as well as".

Reply: We remove 'as well as', thank you.

Line 59: "added it into the basic functions" → "added it to the basic functions"

Reply: thank you, we correct it.

Line 64: "some major climate variability such like the El Niño−Southern Oscillation" → "such as"

Reply: thank you, we change it.

Line 79: "the neural networks" → "neural networks"

Reply: we remove 'the'.

Line 89: "extrapolations on rates" → better: "extrapolations of rates"

Reply: we change it, thank you.

Line 93: "examination for reginal sea level rise" → should be "regional"

Reply: sorry for the typo.

Line 100: "distinguished literatures" → "seminal studies" or "notable publications"

Reply: good suggestion, we change it to 'notable publications'

Line 104: "use to reconstruct"  $\rightarrow$  "use it to reconstruct"

Reply: thank you, we correct it.

2.1 Title: "Sea level reconstruction by data assimilation"  $\rightarrow$  Consider: "Sea level reconstruction using data assimilation"

Reply: thank you for the suggestion, we adopt it.

Line 110: "to facilitate understanding for readers" → redundant; delete or simplify: "to facilitate understanding"

Reply: thank you, we make it concise.

Line 116: "physically orientated" → should be "physically oriented"

Reply: thank you, we correct the typo.

Line 120: "including Greenland ice melting..." → better as "including mass loss from the Greenland Ice Sheet..."

Reply: thank you for the better wordings.

Line 124: "Those oceanic geometries are termed as sea level fingerprint" → "These oceanic patterns are termed sea level fingerprints"

Reply: thank you, we remove 'as'

Line 126: "A random process is further proposed..." → awkward. Try: "We also introduce a random process..."

Reply: thank you for offering a better phrase.

Line 200: "clime model" → "climate model"

Reply: thank you for correcting the typo for us.

Line 215: "we do not exclude records with large jumps or high rates, as their impact... is negligible"  $\rightarrow$  requires justification or citation.

Reply: good suggestion, we quantified it, and you can see our response to your major concern 5 Data Usability.

2.6 GIA description: "mainly is an ongoing response..." → should be "is mainly an ongoing response..."

Reply: thank you for the correction.

Line 254: "see section 'Code and data availability" → inconsistent with other section references; consider standardizing.

Reply: thank you for the suggestion, we refer it to Mu (2025), i.e., the reference describing the data deposited at Zenodo.

2.7: "Reconstruction from literatures" → should be "Reconstructions from previous studies" or "Existing reconstructions"

Reply: we adopt this suggestion, thank you.

Line 265: "exerted broad influence" → more objective phrasing is "widely used"

Reply: thank you for the suggestion.

Table 2 Caption: "Sea level reconstruction from literatures" → "Overview of sea level reconstruction studies"

Reply: thank you for offer a better choice.

2.8: "we average the weekly samples into annual time series..." → passive form might be clearer: "The weekly data were averaged to annual time series..."

Reply: thank you, we change it.

2.9: "we select the nearest grid point from T2024 for each site of tide gauge"

"...for each tide gauge site"

Reply: than you, we change it.

Line 315: "committed to address" → "dedicated to addressing"

Reply: thank you, we adopt the suggestion.

Line 318: "illustrate diverse reconstructions at tide gauges" → redundant phrasing. Better: "illustrate the diversity in reconstructions"

Reply: thank you, we rephrase it.

Line 323: "regardless their durations" → "regardless of their duration"

Reply: thank you, we correct it.

Line 324–326: Repetition of "physically" in "(physically) simulated sea level..." is awkward and unnecessary.

Reply: thank you for the suggestion, we remove them.

Line 332: "tend to converge over periods when raw records are available" → could be shortened: "converge when raw records are available"

Reply: good suggestion, we adopt it.

Line 460: "rate differences are very minor" → better: "rate differences are generally small"

Reply: we change the wordings. Thank you.

Line 470: "sea level rates are expected to be high over a short period" → maybe clarify: "rate estimates are more variable over short periods"

Reply: thank you, we rephrase the wordings.

Line 477: "use of these data should be with cautions" → "should be used with caution"

Reply: thank you, we change it.

Line 480: "anthropologic activities" → "anthropogenic activities"

Reply: thank you for the correction.

Line 485: "not purely relevant to SDSL or SLF changes" → unclear. Better: "not directly attributable to SDSL or SLF mechanisms"

Reply: thank you, we change the wordings.

Line 490: "we did not remove those tide gauges... because, first, we intend to include..." → awkward. Suggest breaking into two sentences and rewriting as: "We retained all gauges to maximize spatial coverage. Moreover, the impact of anomalous records is localized and does not significantly affect other stations."

Reply: Great suggestion, thank you. We rephrase the wordings.

Line 504: "Sea level rates spanning period less than 30 years must be explained with cautions..." → "Sea level rates estimated over periods shorter than 30 years should be interpreted cautiously..."

Reply: thank you for offering a better phrase.

Line 510: "offer an ensemble of complete, refined, and smooth time series" → could be shortened: "provide refined, continuous time series"

Reply: thank you, we adopt this suggestion.

Line 514: "align with sea level observations and other sea level reconstructions..."  $\rightarrow$  redundant use of "sea level"; remove one.

Reply: we remove both.

Line 517: "our reconstructions advocate the raw records of tide gauges" → "closely follow" or "are consistent with" is clearer than "advocate"

Reply: we change it to 'closely follow'

Line 519: "expected to contribute..." → "expected to support efforts to understand..."

Reply: thank you, we adopt this suggestion.

Line 521: "It contains following variables" → "It contains the following variables:"

Reply: we add the word 'the'

Line 530: "missing values are assigned with 'NaN'." → "missing values are denoted by 'NaN'."

Reply: thank you, we adopt this suggestion.

Line 534: "contains the spread of sea level reconstructions..." → maybe clarify: "the ensemble spread (standard deviation) across models"

Reply: thank you, we change it.

Line 539: "scripts are also available upon request to..."  $\rightarrow$  better to specify whether code will be publicly released or must be requested; ESSD encourages transparency.

Reply: At the moment, the code is only available upon request, but we are committed to release an open source code as soon as possible.